# Violence against older women: A systematic review of qualitative literature

**Sarah R. Meyer**[1]*, **Molly E. Lasater**[2], **Claudia García-Moreno**[1]

**1** Department of Sexual and Reproductive Health and Research, World Health Organization, Geneva, Switzerland, **2** Department of Mental Health, Johns Hopkins Bloomberg School of Public Health, Baltimore, Maryland, United States of America

* smeyer@who.int

**Data Availability Statement:** All relevant data are within the manuscript and its Supporting Information files.

**Funding:** This study is funded by the Department for International Development, UNWomen-World

## Abstract

The majority of the existing evidence-base on violence against women focuses on women of reproductive age (15–49), and globally there is sparse evidence concerning patterns of and types of violence against women aged 50 and older. Improved understanding of differing patterns and dynamics of violence older women experienced is needed to ensure appropriate policy or programmatic responses. To address these gaps in the evidence, we conducted a systematic review of qualitative literature on violence against older women, including any form of violence against women, rather than adopting a specific theoretical framework on what types of violence or perpetrators should be included from the outset, and focusing specifically on qualitative studies, to explore the nature and dynamics of violence against older women from the perspective of women. Following pre-planned searches of 11 electronic databases, two authors screened all identified titles, abstracts and relevant full texts for inclusion in the review. We extracted data from 52 manuscripts identified for inclusion, and conducted quality assessment and thematic synthesis from the key findings of the included studies. Results indicated that the vast majority of included studies were conducted in high-income contexts, and did not contain adequate information on study setting and context. Thematic synthesis identified several central themes, including the intersection between ageing and perceptions of, experiences of and response to violence; the centrality of social and gender norms in shaping older women's experiences of violence; the cumulative physical and mental health impact of exposure to lifelong violence, and that specific barriers exist for older women accessing community supports and health services to address violence victimization. Our findings indicated that violence against older women is prevalent and has significant impacts on physical and mental well-being of older women. Implications for policy and programmatic response, as well as future research directions, are highlighted.

## Introduction

Violence against women is a major public health problem, a gender inequality issue and a human rights violation. There are significant serious and long-lasting impacts of violence on women's physical and mental health, including injuries, unintended pregnancy, adverse birth

Health Organization Joint Programme on
Strengthening Methodologies and Measurement
and building national capacities for Violence
against Women data. The funders had no role in
study design, data collection and analysis, decision
to publish, or preparation of the manuscript.

**Competing interests:** The authors have declared
that no competing interests exist.

outcomes, abortion (often in unsafe conditions), HIV and sexually transmitted infections, depression, alcohol-use disorders and other mental health problems [1–5]. The 2030 Sustainable Development Goals [SDGs] include as one of their targets (5.2) under Goal 5 on gender equality, the elimination of all forms of violence against women and girls. Indicator 5.2.1, measuring intimate partner violence [IPV]: Proportion of ever-partnered women and girls aged 15 years and older subjected to physical, sexual or psychological violence by a current or former intimate partner in the previous 12 months, is proposed to track the measurement of progress in achieving this goal. The indicator does not include an upper age limit, and data on older women (aged 50 and above), including but not limited to intimate partner violence, are needed to support national and global monitoring of violence against women of all ages, including for monitoring of the SDGs.

The majority of existing violence against women surveys and data have focused on women of reproductive age (15–49), as they suffer the brunt of intimate partner violence and non-partner sexual violence [6]. A growing number of surveys are now including women older than 49 years, however globally there is sparse evidence concerning patterns of and types of violence against women aged 50 and older, and limited understanding of barriers to reporting and help-seeking amongst older women who are subjected to violence [7]. Compared to women of reproductive age, women aged 50 and above may experience different relationship dynamics which influence forms of abuse [8, 9], and some evidence indicates that older women experience different types of violence, for example, psychological violence and verbal abuse, compared to younger women's experiences of physical and sexual violence [10]. For older women, recent exposure to violence may be interlinked with violence victimization at different stages of the life-course [11, 12]. Dynamics of ageing may shape experiences of violence, for example, provision of care to a dependent partner may influence decisions to disclose or report abuse [10]. They are also more likely to experience violence from other family members, including children, and from carers. Currently, the evidence-base of qualitative and quantitative data concerning violence against older women is limited, and a better understanding of these differing patterns and dynamics is needed to ensure appropriate policy or programmatic responses to violence against older women and service development and provision for older women affected by violence [10, 11]. To address these gaps in the evidence, we conducted a systematic review of qualitative literature on violence against older women.

## Current frameworks on violence against women and existing evidence

Gaps in research and evidence stem in part from conflicting theoretical approaches, definitions and conceptual frameworks concerning violence against older women. The dominant theoretical frameworks are the older adult mistreatment framework and older adult protection framework [7, 13, 14]. The older adult mistreatment framework conceptualizes violence against older women as a form of elder abuse, focusing on age as the primary factor influencing vulnerability to exposure to violence. The older adult protection framework specifically understands violence within the context of care-giving and institutional arrangements, where older adults' often be gender neutral, and the adult protection framework can result in a framing of older adults as inherently impaired and vulnerable. In addition, the IPV framework primarily understands vulnerability to violence in terms of gender inequality and partnership dynamics, which may neglect analysis of how ageing and partner violence intersect. These differing frameworks inform multiple aspects of research, including study design, data collection and analysis, and reporting, resulting in fragmented data and evidence. For example, some research utilizing the older adult mistreatment framework lacks a focus on the gendered dimensions of violence [14, 15], and other studies have solely focused on women in

institutional settings, neglecting measurement of violence perpetrated by intimate partners and other family members [13].

Existing syntheses of evidence on violence against older women often reflect these differing conceptual frameworks. Employing an older adult mistreatment framework, a systematic review of quantitative studies of elder abuse (against men and women aged 60+) found that the global prevalence of elder abuse in community settings is 15.7% in the past year, with psychological abuse and financial abuse as the most prevalent forms of abuse reported [16]. This review reported prevalence by type of violence, but did not report on perpetrators. Analysis of studies conducted in institutional settings found women, aged 60 and above, to be significantly more vulnerable to abuse, with psychological abuse as the most prevalent form of violence, followed by physical violence, neglect, financial and sexual abuse [17]; this analysis included data reporting staff-to-resident abuse. Analysis of quantitative data of women aged 60 and above in the systematic review of quantitative studies of elder abuse found a global prevalence of elder abuse against women of 14.1% in the past year, with psychological abuse reported as the most prevalent form of violence, followed by neglect [16]. The focus of this review was prevalence of different sub-types of violence, and type of perpetrator was not considered. Another systematic review of quantitative data on interpersonal violence (physical and/or sexual violence) against older women (aged 65 and above) in community dwellings primarily employed an IPV framework, finding prevalence of reported interpersonal violence ranged from 6 to 59% over a lifetime, from 6 to 18% since turning 50, and 0.8 to 11% in the past year, however, results indicated that definitions of violence vary widely and affect prevalence estimates [18]. Syntheses of quantitative literature have identified prevalent forms of violence against older women, highlighting limitations in the evidence-base due to variations in definitions and methodology, and a primary emphasis on populations in high-income, Western countries. These reviews have captured a wide range of types of violence, however, have not considered type of perpetrators or patterns of co-occurring types of violence.

Alongside these systematic reviews of quantitative data, some reviews have included qualitative and mixed methods studies. An empirical review of IPV in later life examined 27 quantitative, 22 qualitative and 7 mixed-methods studies, finding that forms of IPV amongst older women in later life shifted from a higher prevalence of physical and sexual abuse during reproductive years, to a higher prevalence of forms of psychological abuse [19]. A review of qualitative research on IPV amongst older women identified a number of relevant themes, including patterns of abuse that were continuous and consistent with previous experiences of abuse in families of origin and previous relationships [20]. A systematic review and meta-synthesis of qualitative studies of IPV and older women focused on how previous exposure to IPV influenced health-seeking behaviours, specifically mental health care [21]. An empirical review of quantitative and qualitative studies of sexual violence against older people identified widespread variation in prevalence rates across studies, and a range of perpetrators, primarily intimate partners or adult children [22]. A recent narrative review of quantitative, qualitative and mixed methods studies of IPV against women aged 45 and above concluded that women's "age and life transitions mean that they may experience abuse differently to younger women. They also face unique barriers to accessing help, such as disability and dependence on their partners" [23].

However, amongst these existing systematic reviews of qualitative literature, none have focused specifically on older women, while also being inclusive of any form of violence. In order to improve understanding of violence against older women, it is important to explore patterns, dynamics and experiences through examination of the qualitative literature. Qualitative data on violence against older women complements quantitative evidence not only by offering insight into lived experiences of older women subjected to violence, but also by

expanding and clarifying types of violence, perpetrators, linkages to particular risk factors, and physical, mental and social impacts of violence against older women.

In the present review, we aimed to build on previous systematic reviews and strengthen the evidence-base by i) including studies and evidence focused specifically on women; ii) including *any* form of violence against women, rather than adopting a specific theoretical framework on what types of violence or perpetrators should be included from the outset; iii) focusing on women aged 50 and above (as many surveys often specifically focus on women of reproductive age, which is considered to be up to 49 years of age); and iv) focusing specifically on qualitative studies, to explore the nature and dynamics of violence against older women from the perspective of women. We aimed to identify, evaluate and synthesize qualitative studies from all countries, exploring violence against women aged 50 and above, identifying types and patterns of violence, perpetrators of violence, and impacts of violence on various outcomes for older women, including physical and mental health and social support, and women's responses to experiences of violence. We include the following forms of violence: elder abuse, family violence and intimate partner violence. Elder abuse is defined as "single or repeated act, or lack of appropriate action, occurring within any relationship where there is an expectation of trust which causes harm or distress to an older person" [24]. Intimate partner violence is defined as "behaviour by an intimate partner or ex-partner that causes physical, sexual or psychological harm, including physical aggression, sexual coercion, psychological abuse and controlling behaviours" [25]. Family violence is often used interchangeable with intimate partner violence, however, also encompasses abuse and violence perpetrated by other family members, for example, adult children or in-laws. While there is no universal agreed-upon definition of older women, for the purposes of this review, we define older women as women aged 50 and above, while recognizing that aging and age are social phenomenon, and definitions vary across organizations, cultures and communities. The protocol was pre-registered with PROSPERO, Registration Number: CRD42019119467, https://www.crd.york.ac.uk/prospero/display_record. php?ID=CRD42019119467 (see also [26]).

## Methods

### Search strategy

In this systematic review, we searched 11 electronic databases–PubMed, PsycINFO, Embase, CINAHL, PILOTS, ERIC, Social Work Abstracts, International Bibliography of the Social Sciences, Social Services Abstracts, ProQuest Criminal Justice and Dissertations & Theses Global, from 1990. We conducted searches that combined the following domains as part of the research question: 1) age (50 and above); AND 2) women; AND 3) violence; AND 4) qualitative methodology. For each of these domains, we identified the relevant keywords and search terms, which varied by database; the search strategy was appropriately modified for each database, including syntax and specific terms, topics and/ or headings. The search strategy for PubMed is included in S1 File. Searches were conducted in April 2018 and updated in July 2019. We did not limit the search by year of publication or language.

We also hand searched reference lists of relevant existing systematic reviews, which we identified both through background research and through the formal database searches, and reviewed relevant references (44 identified). We consulted with 49 experts on violence against older women or older adults, including researchers, practitioners and policy makers, from all regions globally. All experts were contacted and followed-up with a minimum of 2 contacts. 26 experts responded with 424 articles, 64 of which were duplicates. We reviewed the full text of 43 articles and ultimately included 2 in the full review. Grey literature was not systematically searched; grey literature submitted by experts was initially considered for inclusion, however,

conducting comparable data extraction and quality assessment for grey literature alongside the peer-reviewed literature was not possible.

We identified 18 non-English language articles for full-text review. For 17 of these articles, we identified a native speaker external reviewer who was provided with inclusion and exclusion criteria and consulted with authors regarding final inclusion (4 Portuguese, 7 Spanish, 1 Hebrew, 1 Dutch, 1 German, 1 Danish, 2 French). One non-English article (in Farsi) was not reviewed as the research team could not engage a Farsi speaker to review the article. The external reviewers consulted with SRM to decide on inclusion of full texts, and conducted data extraction and quality assessment on 3 articles identified for inclusion (2 Spanish, 1 Portuguese) [27–29].

## Study selection and data extraction

After removing duplicates, study selection proceeded in two stages: in the first stage, two authors (SRM and MEL) reviewed titles and abstracts of all identified manuscripts. We included studies that met the following criteria: i) focused on women aged 50 and older, ii) employed qualitative methodology, and iii) focused on women's experiences of any type of violence perpetrated by any type of perpetrator. Studies including men or also including women aged younger than 50 were included if specific and separate sex and age-specific analyses were included. We included studies employing any type of qualitative methodology, and mixed methods studies were included if qualitative data was presented separately. Studies were excluded if the whole sample was children, adolescents or adults under the age of 50; if the sample only included men; if the methodology was quantitative, or in the case of mixed methods studies, if the qualitative results were not separately presented, and if the data only included the perspectives on violence against women as reported by care providers, health professionals, legal professionals and nursing home managers.

After the first stage of title and abstract review, we reviewed the full text of any manuscript considered relevant by either of the authors. In the second stage, two authors (SRM and MEL) independently reviewed all articles selected for full text review for eligibility, to reach consensus on inclusion. Any discrepancies were resolved with the input of an external reviewer. Fig 1 indicates the full search and study selection process.

We designed a data extraction Excel spreadsheet specifically for the purposes of the review, including characteristics of included studies (location of the research, research question), methodology (conceptual framework or theoretical approach, data collection methods, data analysis methods, sampling), characteristics of the sample (inclusion and exclusion criteria, brief description of the sample), types and nature of violence (context of violence, perpetrator and brief description of impacts of violence). We extracted main findings, participant quotations where possible, and study limitations, if reported. Data extraction was conducted by one author (MEL), and checked for accuracy by a second author (SRM), with discrepancies resolved by discussion to reach consensus.

## Quality assessment

All included studies were assessed for quality using an adapted version of the Critical Appraisal Skills Programme [CASP] scale. The adapted scale included the following questions [30]:

1. Was there a clear statement of the aims of the research?

2. Is a qualitative methodology appropriate?

3. Are the setting(s) and context described adequately?

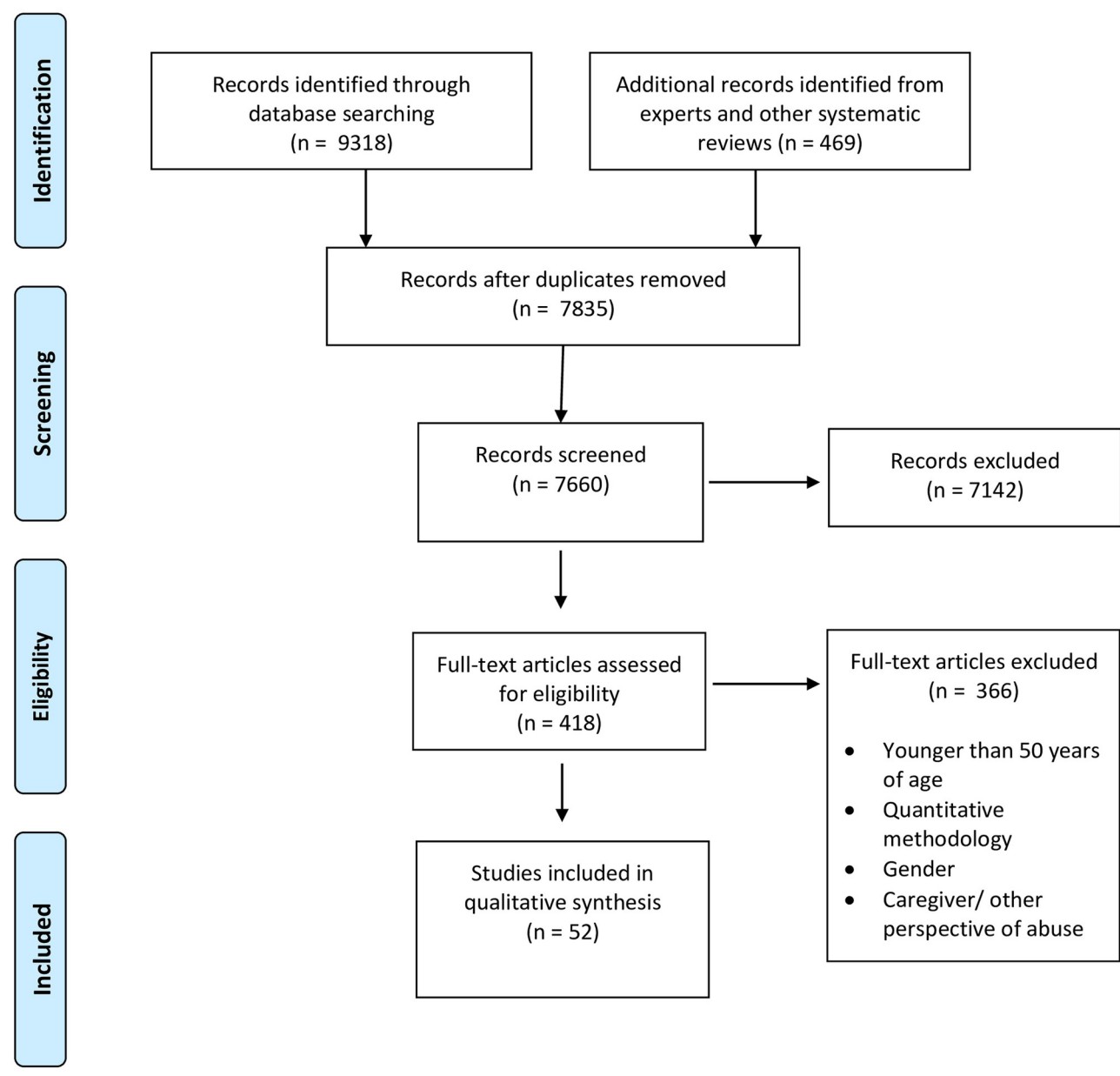

**Fig 1. Identification of included studies.** *From*: Moher D, Liberati A, Tetzlaff J, Altman DG, The PRISMA Group (2009). Preferred Reporting Items for Systematic Reviews and Meta-Analyses: The PRISMA Statement. PLoS Med 6(6): e1000097. doi:10.1371/journal.pmed1000097. For more information, visit www.prisma-statement.org.

4. Was the research design appropriate to address the aims of the research?

5. Is the sampling strategy described, and is this appropriate?

6. Is the data collection strategy described and justified?

7. Is the data analysis described, and is this appropriate?

8. Are the claims made/findings supported by sufficient evidence?

9. Is there evidence of reflexivity?

10. Does the study demonstrate sensitivity to ethical concerns?

Two authors (SRM, MEL) assessed the quality of the studies, assigning a 1 for each affirmative response and 0 for each negative response, for a final score out of 10. Disagreement was resolved by discussion between the two authors. Quality assessment was not used to determine if any studies should be excluded, but rather to assess the strength of each study.

### Synthesis

An Excel spreadsheet to compile all relevant findings and quotations from the studies for thematic analysis was developed. Two of the authors (SRM and MEL) coded the main findings extracted from each study. We used line-by-line coding on a sub-set of articles, developing a set of over-arching themes and sub-themes for a draft codebook. The coding proceeded as an iterative process, with the two authors each separately coding the main findings using the draft codebook, discussing coding results, and refining the codebook based on overlap and redundancies identified. After all data were coded, we tallied all occurrences of each code and further explored areas of overlap and merged sub-themes with low numbers of codes, finalizing the broad themes and focused sub-themes. For non-English articles included, the external reviewer translated primary quotations into English and thematic analysis on these articles was conducted alongside the English language articles.

### Reporting

The synthesis and all aspects of the systematic review process are reported following the 21-item checklist provided in the Enhancing Transparency in Reporting the Synthesis of Qualitative Research statement [31] and the PRISMA checklist [S2 and S3 Files].

## Results

### Studies identified and characteristics

Our searches of 11 databases yielded 9318 articles, with an additional 468 articles identified through cross- referencing and expert recommendation. After removing duplicates, 7834 articles remained. We identified 417 articles that were potentially eligible and included in full text screening. Two of these articles had not yet been published. Additionally, 1 Farsi language study was unable to be translated and assessed against the selection criteria. Fifty-two articles met criteria for inclusion in this systematic review (Fig 1). The 52 included articles represent data from 31 studies.

### Overview of study characteristics

**Study setting (Table 1).** Most studies were conducted in high-income countries (HIC), including the United States of America (n = 16), Israel (n = 12), Canada (n = 7), the United Kingdom (n = 4), Hong Kong (n = 2), and Australia (n = 1). Six articles were from upper-middle income countries–Brazil (n = 3), Mexico (n = 2) and Iran (n = 1); and three articles were from low-income countries–Uganda (n = 1) and Ethiopia (n = 2). One article came from India, a lower-middle income country.

### Quality assessment

Application of the adapted version of CASP scale yielded variable results across the 52 articles assessed [see Table 2]. Ratings of research methodology, statement of research aims and selection of appropriate research design were overall high. The majority (46 articles) [29, 32–76]

**Table 1. Characteristics of included studies.**

| First author | Publication year | Study location–country (region) | Research question(s) | Sample (number, age range) | Data collection method and analysis method | Type(s) of violence and perpetrator(s) |
|---|---|---|---|---|---|---|
| Agoff | 2006 | Mexico (AMRO) | To identify personal, cultural, and institutional factors that hinder resolution of domestic violence, to identify factors that facilitate the violence. | 26; Age range: 26–72 | Open-ended face-to-face interview; analysis guided by grounded theory, focusing on two aspects: subjective perceptions of violence and barriers to overcoming violence | IPV–any type |
| | | | | | | Male partner |
| Ayres | 2001 | United States (AMRO) | How do you define the concept of abuse within the context of ageing women who are at risk for or experiencing physical or emotional injury inflicted by elderly family members for whom they provide care? | 11 | Transcripts of first session of a community-based intervention; concept analysis | Elder abuse–Verbal, physical, emotional abuse |
| | | | | 50 and older for caregiver; 55 and older for elderly family member | | Elderly family member receiving care (spouses or parents) |
| Band-Winterstein | 2009 | Israel (EURO) | How is intimate violence shaped and how does it change throughout the lives of older battered women? How is continuous IPV experienced in old age and how age and violence interact and change throughout the life span | 40 couples; Age range: 60–84 | Face-to-face in-depth interviews; content analysis | IPV–Physical, emotional, economic, psychological |
| | | | | | | Male partner |
| Band-Winterstein | 2010 | Israel (EURO) | What are the unique experiences of old battered women from the dimensions of intentionality of the body in time and space? | 25; Age range: 60–84 | Face-to-face in-depth interviews; content analysis | IPV–physical, emotional |
| | | | | | | Male partners |
| Band-Winterstein | 2010 | Israel (EURO) | What are various perceptions of the attempts to forgive others and the self throughout this lifelong process, as described by older women who have lived with intimate partner violence? What are the lived experiences of forgiveness of older abused women throughout a life of IPV? | 21; Age range: 60–80 | Face-to-face semi-structured in-depth interviews; content analysis | IPV–physical, emotional |
| | | | | | | Male partners |
| Band-Winterstein | 2012 | Israel (EURO) | Explore the constructions of aging in intimate partner violence as narratives of couplehood or narratives of old age; explore how couples, who are living in lifelong IPV, constructed aging in IPV. | N = 30 (15 couples, n = 15 women) | Face-to-face in-depth interviews; dyadic analysis approach focused on identifying overlap and contrast in the couple data; analyzed transcripts as whole life story, then performed separate categorical-content analysis consistent with the narrative approach. | IPV–physical, sexual |
| | | | | Male partner | | Age range: 62–84 |
| Band-Winterstein | 2014 | Israel (EURO) | How do parents experience their aging process in the context of being abused by their adult children with mental disorder? How do they describe the influence of the aging process on the relationship dynamics? How does living in such shared reality impacts their aging needs? | 16 parents (11 mothers); Age range: 58–94 | Face-to-face in-depth interviews; content analysis in phenomenological method | Family violence—physical, emotional abuse, financial, neglect |
| | | | | | | Adult children with mental disorders |

*(Continued)*

**Table 1.** (Continued)

| First author | Publication year | Study location–country (region) | Research question(s) | Sample (number, age range) | Data collection method and analysis method | Type(s) of violence and perpetrator(s) |
|---|---|---|---|---|---|---|
| Band-Winterstein | 2015 | Israel (EURO) | What is the lived experience of elderly women with lifelong IPV? | 31<br><br>Age range: 60–84 | Face-to-face in-depth interviews; phenomenological analysis | IPV–physical, sexual, emotional, economic<br><br>Male partner |
| Band-Winterstein | 2015 | Israel (EURO) | What are the subjective experiences of family members involved in violent, abusive, and neglecting relationships?<br><br>What is an abusive relationship? What does it mean to suffer? What are the perceptions of those who are being abused? What are the elements that make life in abuse possible? How do actors involved in the drama of abuse give coherence of their life? | 11 dyads (parent and child); Age range of parents: 65–90 | Face-to-face in-depth interviews; thematic analysis–identifying the basic components of the experience and placing them into units of meaning according to the study aim, coding and conceptualizing into unique theoretical categories, and organizing main themes and describing the reciprocal relations between them | Family violence/ elder abuse–physical violence, verbal aggression, financial exploitation, and forms of neglect.<br><br>Child |
| Band-Winterstein | 2019 | Israel (EURO) | To differentiate between the lived experience of two groups of women caregiving for a partner with dementia; One group was coping with lifelong IPV and dementia-related violence (Group 1); the other group was coping with dementia-related violence only (Group 2). | 16; Age range: 63–84 | In-depth, semi-structured face-to-face phenomenological interviews; Interpretive phenomenological analysis (IPA) | IPV–physical, sexual, verbal<br><br>Male partner |
| Barbosa | 2015 | Brazil (AMRO) | To understand the impact of sexual violence suffered by women with mental disorders based on self-reports of these experiences. | 17; Age range: 18–68 | Face-to-face in-depth interviews; structured narration analysis | IPV–sexual, physical |
| Bhatia | 2019 | India (SEARO) | To unearth the causes of partner violence in later life, to understand the patterns of partner violence in later life and to understand psychological and social consequences faced by women undergoing partner violence in later life. | 38; Age range: 50 and above | Face-to-face in-depth interviews (4) and focus group discussions (2); Analysis methods not described | IPV and family violence–physical, emotional, financial<br><br>Husband, male partner, other relatives |
| Buchbinder | 2003 | Israel (EURO) | Describe and analyze the experiences and perceptions of older battered women in coping with and surviving the violence. | 20; Age range: 60–80 | Face-to-face in-depth interviews; content analysis in phenomenological method | IPV–physical, psychological, and sexual<br><br>Male partner |
| Chane | 2015 | Ethiopia (AFRO) | What is the lived experience of abused elders and how can we increase understanding of elder abuse? | 15 (9 women); Age range: 64–93 | Face-to-face in-depth interviews; interpretative phenomenological analysis informed by hermeneutic phenomenology | Family violence/ elder abuse–financial, physical, psychological<br><br>Family members, community members |
| Chane | 2015 | Ethiopia (AFRO) | What are the types and nature of abuse and neglect from the perspective of elders in Ethiopia who experienced abuse in noninstitutional settings? | 15 total, 9 women; 64–85 | Face-to-face in-depth interviews; coding following interpretative phenomenological analysis approach | Family violence/ elder abuse–financial, physical, psychological<br><br>Family members, community members |

*(Continued)*

**Table 1.** (*Continued*)

| First author | Publication year | Study location–country (region) | Research question(s) | Sample (number, age range) | Data collection method and analysis method | Type(s) of violence and perpetrator(s) |
|---|---|---|---|---|---|---|
| Cheung | 2015 | Hong Kong (WPRO) | How does IPV victimization manifest itself among older women? | 2; 63 and 69 | Not described; not described | IPV–Verbal, physical, controlling behaviours, financial, emotional |
| | | | | | | Male partners |
| Cronin | 2013 | USA (AMRO) | How do women make meaning with their experiences with domestic violence; The focus of this study is women's lives after violence, and the ways in which they have coped with the challenges of living and aging. | 15; Age range: 60–89 | Face-to-face in-depth interviews; narrative life history approach to coding | IPV–Physical controlling behaviours, verbal, emotional, financial control |
| | | | | | | Male partners |
| de Menezes | 2013 | Brazil (AMRO) | To analyze the aggressive behavior in the relationship between elderly with symptoms of dementia and their family caregivers. | 4 couples–each pair aggressor and caregiver; Age range of caregivers: 68–77 | Semi-structured interviews; thematic content analysis | IPV and family violence–physical, threats, psychological |
| | | | | | | Elderly receiving care |
| Eisikovits | 2015 | Israel (EURO) | What are the ways in which young and old battered women perceive, understand and experience suffering from violence, how do they build these experiences into the central theme of their life and how do they reconstruct them in a manner that makes their lives livable? | 17; Age range: 60–84 | Semi-structured in-depth interviews; content analysis | IPV–physical, psychological |
| | | | | | | Male partners, husbands |
| Fakari | 2013 | Iran (EMRO) | Describe daily life experience (of violence against older women) just in the same way they occurred in reality. | 13; mean age 62 | Face-to-face in-depth interviews; "holistic methods of analysis" | IPV and elder abuse–physical and psychological, financial exploitation |
| | | | | | | Not stated |
| Grunfeld | 1996 | Canada (AMRO) | How does violence impact the lives of elderly women? | 4; Age range: 63–73 | Face-to-face open-ended in-depth interviews; thematic analysis | IPV and family violence–physical, emotional, financial, controlling behaviours |
| | | | | | | Husbands, children and grandchildren |
| Guruge | 2010 | Canada (AMRO) | What are older immigrant women's experiences and responses to abuse and neglect? | 43; Age range: 48–85 | In-depth interviews and focus group discussions | IPV and family violence–emotional, physical, sexual, financial abuse, neglect, controlling behaviours |
| | | | | | | Husbands, children, children-in-law |
| Hightower | 2006 | Canada (AMRO) | What is the experience of violence and abuse of women aged 50 and older? | 64; Age range: 50–87 | Interviews and group sessions; not described | IPV and family violence–financial, sexual, physical, emotional/ psychological, controlling behaviours |
| | | | | | | Male partners and other family members |

(*Continued*)

**Table 1.** (Continued)

| First author | Publication year | Study location– country (region) | Research question(s) | Sample (number, age range) | Data collection method and analysis method | Type(s) of violence and perpetrator(s) |
|---|---|---|---|---|---|---|
| Lazenbatt | 2013 | UK (EURO) | How do older women with an abusive partner for more than 30 years cope with domestic violence and how does it affect their wellbeing? | 18; Age range: 53–72 | Face-to-face semi-structured in-depth interviews; thematic analysis | IPV–physical, psychological, controlling behaviours<br><br>Male partner |
| Lazenbatt | 2014 | UK (EURO) | How 'older women' cope with domestic violence and how it affects their wellbeing, using a theoretical framework of 'salutogenesis' to consider coping resources used in lifelong abuse | 18; Age range: 53–72 | Face-to-face in-depth interviews; thematic framework analysis based on 'salutogenesis' theoretical dimensions were used to explore their 'wellbeing and coping' | IPV–physical, psychological/ emotional abuse, sexual abuse, financial exploitation<br><br>Male partner |
| Lichtenstein | 2009 | United States (AMRO) | To identify barriers to reporting domestic violence to law enforcement among older African American women in the rural south. How does age, ethnicity, and gender intersect with rurality and systems such as old boys' networks in creating barriers to reporting domestic violence to law enforcement? | 15; Age range: 50–84 | Focus group discussions (2); constant comparison method | IPV–physical, verbal<br><br>Husband |
| Lowenstein | 1999 | Israel (EURO) | To describe possible reasons for the phenomenon of elder spousal abuse in second marriages, and to identify possible risk factors for abuse based on reports by remarried elderly who were victims of spousal abuse | 12 couples, of which 9 of the women were victims of spousal abuse; 60+ | Face-to-face in-depth interviews; coding–not described further | IPV–physical, controlling/ psychological<br><br>Partners |
| McGarry | 2010 | United Kingdom (EURO) | What are women's experiences of domestic abuse and what is its effect on their health and lives? | 16; Age range: 59–84 | Face-to-face in-depth interviews; iterative approach and informed by the analytic hierarchy model | IPV–physical, emotional, sexual, Male partner |
| McGarry | 2014 | United Kingdom (EURO) | What are the service responses to abuse among older people across a range of sectors? What are the perspectives of older people either as survivors of abuse or as older people with an interest in service development? | 3; Age range: 60–65 | Semi-structured phone interviews; Analytic Hierarchy Mode and constant comparative method | Elder abuse, family violence and IPV<br><br>Any |
| Montminy | 2005 | Canada (AMRO) | How is psychological violence against older women experienced in the marital context? | 15; Age range: 60–81 | Face-to-face in-depth interviews; manifest content analysis | IPV–psychological<br><br>Male partner |
| Nahmiash | 2004 | Canada (AMRO) | What is the interacting relationship between the environmental context of care giving and abuse and neglect of older adults. | 16 participants (14 victims, 2 abusers); 12 of the 14 victims were female; Age range: 61–78 | Face-to-face in-depth interviews; content analysis | Elder abuse–sexual, physical<br><br>Care-givers and/ or partners |

(*Continued*)

**Table 1.** (Continued)

| First author | Publication year | Study location–country (region) | Research question(s) | Sample (number, age range) | Data collection method and analysis method | Type(s) of violence and perpetrator(s) |
|---|---|---|---|---|---|---|
| Pillemer | 2011 | USA (AMRO) | What are the major forms of resident to resident aggression that occur in nursing homes? | 53 units in 3 facilities, 122 events identified; no age range specified | Identified all resident-to-resident aggression events in several nursing homes over 2 week period through resident interview, certified nursing assistant interview, and interviewer observation; sorted events into categories | Elder abuse<br><br>Other residents or nursing homes |
| Ramsey-Klawsnik | 2003 | USA (AMRO) | What are the patterns of elder sexual abuse, both marital and incestuous? What are the abuse dynamics, problems confronting victims, and perpetrator characteristics? | 130 cases (consultation files); not specified | Review of consultation files from Protective Services Program of the Massachusetts Executive Office of Elder Affairs; analysis method not described beyond "qualitatively analysed" | IPV and elder abuse–sexual<br><br>Partner, caregiver, family members |
| Richards | 2013 | Uganda (AFRO) | How women's and men's gendered experiences from childhood to old age have shaped their vulnerability in relation to HIV both in terms of their individual risk of HIV and their access to and experiences of HIV services | Total 31; 16 women. Age range: 60 and over | Face-to-face in-depth interviews and FGDs; framework approach to coding | IPV–physical, sexual and psychological<br><br>Male partners |
| Roberto | 2013 | USA (AMRO) | What are the issues facing rural older women who wish to lead safe and violence-free lives and to identify the com-munity support needed to help them successfully rebuild their lives. | 10; Age range: 54–70 | Face-to-face in-depth interviews; not described | IPV–emotional, physical and sexual<br><br>Male partners |
| Roberto | 2018 | USA (AMRO) | How women experienced IPV over the course of their lives and in different contexts; what resources were helpful when older women exited abusive partnerships | 10; Age range: 54–70 | Face-to-face in-depth interviews; open coding and focused coding | IPV–Emotional, physical, financial exploitation<br><br>Male partners |
| Ron | 1999 | Israel (EURO) | What are the main factors, particularly social factors such as the need for intimacy and sexuality, which cause tension among elderly remarried couples and lead to abuse by the spouses? | 12 couples, of which 9 of the women were victims of spousal abuse; 60+ | Face-to-face in-depth interviews; coding–not described further | IPV–verbal, emotional, physical, financial exploitation, caregiver neglect<br><br>Partner |
| Rosen | 2019 | United States (AMRO) | To analyze legal records to describe in detail acute precipitants of physical elder abuse. | 87 cases; Age range: 60–95 | Analyzed narratives from police reports of acute physical elder abuse; cross-case analysis of narratives in police reports to identify codes, coded narratives | IPV and family violence–physical<br><br>Child, spouse/companion, grand child |
| Ruelas-Gonzalez | 2014 | Mexico (AMRO) | To analyze health care providers and older patients' perceptions about elder abuse by health personnel of public health services. | 6 older women; Age range: 65–87 | Semi-structured interviews; analysis using grounded theory approach, content analysis. | Elder abuse–neglect, psychological violence, discrimination<br><br>Health care professionals and caregivers |

(*Continued*)

**Table 1.** (Continued)

| First author | Publication year | Study location–country (region) | Research question(s) | Sample (number, age range) | Data collection method and analysis method | Type(s) of violence and perpetrator(s) |
|---|---|---|---|---|---|---|
| Schaffer | 2008 | Australia (WPRO) | What are the needs of older and isolated women who live with domestic violence? | 90; Age range: 50–78 | Phone-in–asked women to call in and tell their stories to a nation-wide call in service; some "personal" and "group" interviews; analysis method not described | IPV–type(s) not specified

Male partner |
| Sawin | 2011 | USA (AMRO) | What are the experiences of older women diagnosed with breast cancer while experiencing intimate partner abuse? | 11; Age range: 51–84 | Face-to-face in-depth interviews; coding following hermeneutic phenomenological strategy of inquiry | IPV–financial control, psychological control

Male partner |
| Smith | 2015 | USA (AMRO) | How older women/mothers understand and respond to their adult children who are abusive and/or "difficult"; How older low-income women make sense of their adult children's problems. | 15; Age Range: 62 and older | Face-to-face in-depth interviews; coding (type not specified) | Family violence–disrespect, physical and psychological

Adult child |
| Souto | 2015 | Brazil (AMRO) | What are older Brazilian women's experiences of psychological domestic violence? How do older Brazilian women experience their daily life when they are victims of psychological domestic violence? How do older Brazilian women respond to psychological domestic violence? What are older Brazilian women's needs, expectations, and aims in dealing with the psychological domestic violence in their lives? | 11; Age range: 66–85 | Face-to-face in-depth interviews; Schutz's motivation theory used as framework for thematic coding | Family violence and IPV–psychological violence, including verbal abuse, financial abuse, neglect

Male partner, family members |
| Souto | 2019 | Canada (AMRO) | How is IPV experienced by Portuguese-speaking older immigrant women? How is women's daily life related to IPV? How does this group respond to IPV situations? What are these women's needs, expectations, and aims in dealing with IPV? | 10; Age range: 60–81 | Face-to-face in-depth interviews; Schutz's motivation theory used as framework for thematic coding | IPV–physical, sexual, emotional, economic abuse, controlling behaviours

Male partner |
| Spencer | 2019 | Canada (AMRO) | How family carers of persons with cognitive impairment respond to fear, intimidation, and violence, over time and across different settings | 10; Age range: 23–83, median age 64 (only results attributed to women aged 50 and above included in review analysis) | Participants kept weekly diary of interactions with person for who they provided care, and follow-up interview following completion of diary; narrative analysis | IPV and family violence–physical, verbal, emotional

Husband with dementia (7); mother (3) |

(*Continued*)

**Table 1.** (Continued)

| First author | Publication year | Study location– country (region) | Research question(s) | Sample (number, age range) | Data collection method and analysis method | Type(s) of violence and perpetrator(s) |
|---|---|---|---|---|---|---|
| Teaster | 2006 | USA (AMRO) | What is the trajectory of, and community responses to, violence in late life? Aim is to further understanding of IPV in rural communities by examining responses to violence from the perspective of aging women, as well as those entities intervening in their cases (e.g., APS caseworkers, women's shelters, law enforcement). | 10: Age range: 50–69 | Face-to-face in-depth interviews; open coding and then applied coding scheme developed | IPV–controlling behaviours, physical, verbal, emotional<br><br>Male partners |
| Tetterton | 2011 | USA (AMRO) | What are effective interventions for women above the age of 60 who have experienced IPV? What are the experiences of older women who experienced IPV? | 1; Age range: 63–65 | Face-to-face in-depth interviews; generated case studies from data and used phenomenological approach to conduct thematic analysis | IPV and family violence–physical, emotional<br><br>Male partner and adult son |
| Yan | 2015 | Hong Kong (WPRO) | What are the factors associated with help-seeking behaviors among mistreated elders in Hong Kong? | 40 total, 26 women; Age range: 60–81 | Face-to-face in-depth interviews; grounded theory approach to coding | IPV and family violence–physical, psychological, neglect, financial exploitation, sexual<br><br>Partner, family members |
| Zink | 2003 | USA (AMRO) | What are older women's reasons for remaining in abusive relationships? | 36; Age range: 55–90 | Interviews–some face-to-face, some on telephone; coded using thematic analysis techniques | IPV–physical, emotional, sexual, financial abuse<br><br>Male partner |
| Zink | 2004 | USA (AMRO) | What are the experiences and needs of older victims of IPV in the health care setting? | 38; Age range: 55–90 | Interviews–some face-to-face, some on telephone; coded using immersion crystallization technique | IPV–physical, emotional, financial, sexual<br><br>Male partner |
| Zink | 2006 | USA (AMRO) | What are: (a) the types of abuse perpetrated by older men against their spouses or dating partners and (b) the victim's interpretation of these experiences and behaviors? | 38; Age range: 54–90 | Interviews–some face-to-face, some on telephone; coded using thematic analysis techniques | IPV–physical, emotional, sexual, financial abuse<br><br>Male partner |
| Zink | 2006 | USA (AMRO) | How older women cope in long-term abusive intimate relationships. | 38; Age range: 55–90 | Interviews–some face-to-face, some on telephone; adapted form of grounded theory | IPV–physical, emotional, verbal<br><br>Husband, boyfriend, partner |

gave support for research findings with references to primary data (participant quotations, case study vignettes, case file excerpts). Ten articles [41, 46, 49, 50, 59, 65, 77–80] lacked data analysis descriptions.

Only 12 articles [29, 35, 38–40, 45, 51, 58, 72–74, 79] reflected on the relationship between the researchers and the participants (reflexivity). Procedures for ethical research were described in 36 articles [27, 29, 33–37, 39–41, 45–52, 54, 55, 58, 60, 61, 63, 64, 66–72, 74, 76, 77, 79]. Five articles [43, 44, 56, 73, 75] described obtaining consent, but lacked descriptions of ethical approval, and 10 articles [32, 38, 42, 53, 57, 59, 62, 65, 78, 80] lacked descriptions of both ethical approval and obtaining consent. A significant number of articles [32, 34–42,

**Table 2.**

| Title/ author | Clear statement of research aims? | Appropriate qualitative methodology? | Description of setting and context? | Appropriate research design to address research aims? | Recruitment and sampling strategy is described and appropriate? | Data collection strategy described and justified? | Data analysis described and appropriate? | Findings supported by sufficient evidence? | Evidence of reflexivity? | Ethical issues taken into consideration? | Total score |
|---|---|---|---|---|---|---|---|---|---|---|---|
| Agoff, C., Rajsbaum, A., & Herrera, C. (2006). Perspectivas de las mujeres maltratadas sobre la violencia de pareja en México. *Salud pública de México, 48*(S2), 307–314. | yes | yes | yes | yes | yes | yes | yes | yes | yes | yes | 10 |
| Ayres, M. M., & Woodtli, A. (2001). Concept analysis: abuse of ageing caregivers by elderly care recipients. Journal of Advanced Nursing, 35(3), 326–334. | yes | yes | no | no | no | no | yes | yes | no | no | 4 |
| Band-Winterstein, T., & Avieli, H. (2019). Women Coping With a Partner's Dementia-Related Violence: A Qualitative Study. Journal of nursing scholarship. | yes | yes | no | yes | yes | yes | yes | yes | yes | yes | 9 |
| Band-Winterstein, T., & Eisikovits, Z. (2009). "Aging out" of violence: The multiple faces of intimate violence over the life span. Qualitative Health Research, 19(2), 164–180. | yes | yes | no | yes | yes | yes | yes | yes | no | yes | 8 |
| Band-Winterstein, T., & Eisikovits, Z. (2010). Towards phenomenological theorizing about old women abuse. Ageing International, 35(3), 202–214. | yes | yes | no | yes | no | yes | yes | yes | no | yes | 7 |
| Band-Winterstein, T., Eisikovits, Z., & Koren, C. (2011). Between remembering and forgetting: The experience of forgiveness among older abused women. *Qualitative Social Work, 10*(4), 451–466. | yes | yes | no | yes | no | yes | yes | yes | yes | no | 7 |

*(Continued)*

Violence against older women: a systematic review of qualitative literature

**Table 2.** (Continued)

| Title/ author | Clear statement of research aims? | Appropriate qualitative methodology? | Description of setting and context? | Appropriate research design to address research aims? | Recruitment and sampling strategy is described and appropriate? | Data collection strategy described and justified? | Data analysis described and appropriate? | Findings supported by sufficient evidence? | Evidence of reflexivity? | Ethical issues taken into consideration? | Total score |
|---|---|---|---|---|---|---|---|---|---|---|---|
| Band-Winterstein, T. (2012). Narratives of aging in intimate partner violence: The double lens of violence and old age. Journal of Aging studies, 26(4), 504–514. | yes | yes | no | yes | yes | yes | yes | yes | no | yes | 8 |
| Band-Winterstein, T., Smeloy, Y., & Avieli, H. (2014). Shared reality of the abusive and the vulnerable: The experience of aging for parents living with abusive adult children coping with mental disorder. International Psychogeriatrics, 26(11), 1917–1927. | yes | yes | no | yes | yes | yes | yes | yes | yes | yes | 9 |
| Band-Winterstein, T. (2015). Aging in the shadow of violence: A phenomenological conceptual framework for understanding elderly women who experienced lifelong IPV. Journal of Elder Abuse & Neglect, 27 (4–5), 303–327. | yes | yes | no | yes | yes | yes | yes | yes | no | yes | 8 |
| Band-Winterstein, T. (2015). Whose suffering is this? Narratives of adult children and parents in long-term abusive relationships. Journal of Family Violence, 30(2), 123–133. | yes | yes | yes | yes | yes | yes | yes | yes | yes | yes | 10 |
| Barbosa, J. A. G., Souza, M. C. M. R. D., & Freitas, M. I. D. F. (2015). Violência sexual: narrativas de mulheres com transtornos mentais no Brasil. Revista Panamericana de Salud Pública, 37, 273–278. | yes | yes | yes | yes | no | yes | yes | no | no | yes | 7 |

*(Continued)*

16 / 43

**Table 2.** (Continued)

| Title/ author | Clear statement of research aims? | Appropriate qualitative methodology? | Description of setting and context? | Appropriate research design to address research aims? | Recruitment and sampling strategy is described and appropriate? | Data collection strategy described and justified? | Data analysis described and appropriate? | Findings supported by sufficient evidence? | Evidence of reflexivity? | Ethical issues taken into consideration? | Total score |
|---|---|---|---|---|---|---|---|---|---|---|---|
| Bhatia, P., & Soletti, A. B. (2019). Hushed Voices: Views and Experiences of Older Women on Partner Abuse in Later Life. Ageing International, 44 (1), 41–56. | yes | yes | no | yes | no | yes | no | yes | no | yes | 6 |
| Buchbinder, E., & Winterstein, T. (2003). "Like a wounded bird": Older battered women's life experiences with intimate violence. Journal of Elder Abuse & Neglect, 15(2), 23–44. | yes | yes | no | yes | no | yes | yes | yes | no | no | 6 |
| Chane, S., & Adamek, M. E. (2015). Factors contributing to elder abuse in Ethiopia. The Journal of Adult Protection, 17(2), 99–110. | yes | yes | yes | yes | yes | yes | yes | yes | no | no | 8 |
| Chane, S., & Adamek, M. E. (2015). "Death Is Better Than Misery" Elders' Accounts of Abuse and Neglect in Ethiopia. The International Journal of Aging and Human Development, 82(1), 54–78. | yes | yes | yes | yes | yes | yes | yes | yes | no | no | 8 |
| Cheung, D. S. T., Tiwari, A., & Wang, A. X. M. (2015). Intimate partner violence in late life: a case study of older Chinese women. Journal of Elder Abuse & Neglect, 27(4–5), 428–437. | yes | yes | no | no | no | no | no | no | no | yes | 3 |
| Cronin, V. L. (2013). Silence Is Golden: Older Women's Voices and The Analysis of Meaning Among Survivor's of Domestic Violence. Syracuse University, Dissertation. | yes | yes | no | yes | yes | yes | yes | yes | yes | yes | 9 |

(*Continued*)

**Table 2.** (Continued)

| Title/ author | Clear statement of research aims? | Appropriate qualitative methodology? | Description of setting and context? | Appropriate research design to address research aims? | Recruitment and sampling strategy is described and appropriate? | Data collection strategy described and justified? | Data analysis described and appropriate? | Findings supported by sufficient evidence? | Evidence of reflexivity? | Ethical issues taken into consideration? | Total score |
|---|---|---|---|---|---|---|---|---|---|---|---|
| do Rosário de Menezes, M., Bastos Alves, M., dos Santos Souza, A., Almeida da Silva, V., Nunes da Silva, E., & Souza Oliveira, C. M. (2013). Aggressive Behavior in the relationship between old and the family caregiver in dementias. Ciencia, Cuidado e Saude, 12(4). | yes | yes | no | yes | yes | yes | no | yes | no | yes | 7 |
| Eisikovits, Z., & Band-Winterstein, T. (2015). Dimensions of suffering among old and young battered women. Journal of Family Violence, 30(1), 49–62. | yes | yes | no | yes | yes | yes | yes | yes | no | yes | 8 |
| Fakari, F. R., Hashemi, M. A., & Fakari, F. R. (2013). A Qualitative research: Postmenopausal women's experiences of abuse. Procedia-Social and Behavioral Sciences, 82, 57–60. **R: 1050** | yes | yes | no | no | no | no | no | no | no | no | 2 |
| Grunfeld, A. F., Larsson, D. M., MacKay, K., & Hotch, D. (1996). Domestic violence against elderly women. Canadian Family Physician, 42, 1485. | yes | yes | yes | yes | yes | yes | yes | yes | no | yes | 9 |
| Guruge, S., Kanthasamy, P., Kokarasa, J., Wan, T. Y.W., Chinichian, M. Shirpak, K. R. (2010). Older women speak about abuse & neglect in the post-migration context. Women's Health and Urban Life, 9(2), 15–41. | yes | yes | yes | yes | no | yes | yes | yes | no | yes | 8 |

(Continued)

**Table 2.** (Continued)

| Title/author | Clear statement of research aims? | Appropriate qualitative methodology? | Description of setting and context? | Appropriate research design to address research aims? | Recruitment and sampling strategy is described and appropriate? | Data collection strategy described and justified? | Data analysis described and appropriate? | Findings supported by sufficient evidence? | Evidence of reflexivity? | Ethical issues taken into consideration? | Total score |
|---|---|---|---|---|---|---|---|---|---|---|---|
| Hightower, J., Smith, M. J., & Hightower, H. C. (2006). Hearing the voices of abused older women. Journal of Gerontological Social Work, 46(3–4), 205–227. | no | yes | no | no | yes | no | no | yes | no | yes | 4 |
| Lazenbatt, A., & Devaney, J. (2014). Older women living with domestic violence: coping resources and mental health and wellbeing. Current nursing journal, 1(1), 10–22. | yes | yes | no | yes | yes | yes | no | yes | no | yes | 7 |
| Lazenbatt, A., Devaney, J., & Gildea, A. (2013). Older women living and coping with domestic violence. Community practitioner, 86(2), 28–33. | yes | yes | no | yes | yes | yes | yes | yes | yes | yes | 9 |
| Lichtenstein, B., & Johnson, I. M. (2009). Older African American women and barriers to reporting domestic violence to law enforcement in the rural deep south. Women & Criminal Justice, 19(4), 286–305. | yes | yes | yes | yes | yes | yes | yes | yes | no | yes | 9 |
| Lowenstein, A., & Ron, P. (1999). Tension and conflict factors in second marriages as causes of abuse between elderly spouses. Journal of Elder Abuse & Neglect, 11(1), 23–45. | yes | yes | no | yes | no | yes | yes | yes | no | no | 6 |
| McGarry, J., & Simpson, C. (2010). How domestic abuse affects the wellbeing of older women. Nursing Older People, 22(5), 33–38. | yes | yes | no | yes | yes | no | yes | yes | no | yes | 7 |

(Continued)

**Table 2.** (Continued)

| Title/ author | Clear statement of research aims? | Appropriate qualitative methodology? | Description of setting and context? | Appropriate research design to address research aims? | Recruitment and sampling strategy is described and appropriate? | Data collection strategy described and justified? | Data analysis described and appropriate? | Findings supported by sufficient evidence? | Evidence of reflexivity? | Ethical issues taken into consideration? | Total score |
|---|---|---|---|---|---|---|---|---|---|---|---|
| McGarry, J., Simpson, C., & Hinsliff-Smith, K. (2014). An exploration of service responses to domestic abuse among older people: findings from one region of the UK. The Journal of Adult Protection, 16(4), 202–212. | yes | yes | yes | yes | yes | yes | yes | yes | no | yes | 9 |
| Montminy, L. (2005). Older women's experiences of psychological violence in their marital relationships. Journal of Gerontological Social Work, 46(2), 3–22. | yes | yes | no | yes | yes | yes | yes | yes | no | no | 7 |
| Nahmiash, D. (2004) Powerlessness and Abuse and Neglect of Older Adults. Journal of Elder Abuse and Neglect, 14:1, 21–47. | yes | yes | no | yes | yes | no | yes | yes | no | no | 6 |
| Pillemer, K., Chen, E. K., Van Haitsma, K. S., Teresi, J., Ramirez, M., Silver, S., . . . & Lachs, M. S. (2011). Resident-to-resident aggression in nursing homes: Results from a qualitative event reconstruction study. The Gerontologist, 52(1), 24–33. | yes | yes | yes | yes | yes | yes | yes | yes | yes | yes | 10 |
| Ramsey-Klawsnik, H. (2004). Elder sexual abuse within the family. Journal of Elder Abuse & Neglect, 15(1), 43–58. | no | yes | no | no | no | no | no | yes | no | no | 2 |

(*Continued*)

 

**Table 2.** (Continued)

| Title/ author | Clear statement of research aims? | Appropriate qualitative methodology? | Description of setting and context? | Appropriate research design to address research aims? | Recruitment and sampling strategy is described and appropriate? | Data collection strategy described and justified? | Data analysis described and appropriate? | Findings supported by sufficient evidence? | Evidence of reflexivity? | Ethical issues taken into consideration? | Total score |
|---|---|---|---|---|---|---|---|---|---|---|---|
| Richards, E., Zalwango, F., Seeley, J., Scholten, F., & Theobald, S. (2013). Neglected older women and men: Exploring age and gender as structural drivers of HIV among people aged over 60 in Uganda. African journal of AIDS research, 12(2), 71–78. | yes | yes | yes | yes | yes | yes | yes | yes | no | yes | 9 |
| Roberto, K. A., Brossoie, N., McPherson, M. C., Pulsifer, M. B., & Brown, P. N. (2013). Violence against rural older women: Promoting community awareness and action. Australasian journal on ageing, 32(1), 2–7. | yes | yes | yes | yes | yes | yes | no | no | no | no | 6 |
| Roberto, K. A., & McCann, B. R. (2018). Violence and abuse in rural older women's lives: a life course perspective. Journal of interpersonal violence. | yes | yes | no | yes | yes | yes | yes | yes | no | yes | 8 |
| Ron, P., & Lowenstein, A. (1999). Loneliness and Unmet Needs of Intimacy and Sexuality—Their Effect on the Phenomenon of Spousal Abuse in Second Marriages of the Widowed Elderly. Journal of Divorce & Remarriage, 31(3–4), 69–89. | yes | yes | no | yes | no | yes | yes | yes | no | no | 6 |

*(Continued)*

Table 2. (Continued)

| Title/ author | Clear statement of research aims? | Appropriate qualitative methodology? | Description of setting and context? | Appropriate research design to address research aims? | Recruitment and sampling strategy is described and appropriate? | Data collection strategy described and justified? | Data analysis described and appropriate? | Findings supported by sufficient evidence? | Evidence of reflexivity? | Ethical issues taken into consideration? | Total score |
|---|---|---|---|---|---|---|---|---|---|---|---|
| Rosen, T., Bloemen, E. M., LoFaso, V. M., Clark, S., Flomenbaum, N. E., Breckman, R., . . . Pillemer, K. (2019). Acute precipitants of physical elder abuse: qualitative analysis of legal records from highly adjudicated cases. Journal of Interpersonal Violence, 34(12), 2599–2623. | yes | yes | no | yes | no | yes | yes | yes | no | yes | 7 |
| Ruelas-González, M. G., Pelcastre-Villafuerte, B. E., & Reyes-Morales, H. (2014). Maltrato institucional hacia el adulto mayor: percepciones del prestador de servicios de salud y de los ancianos. salud pública de méxico, 56(6), 631–637. | yes | yes | yes | yes | yes | yes | yes | no | no | no | 7 |
| Sawin, E. M., & Parker, B. (2011). "If looks would kill then I would be dead": intimate partner abuse and breast cancer in older women. Journal of Gerontological Nursing, 37(7), 26–35. | yes | yes | no | yes | yes | yes | yes | yes | no | yes | 8 |
| Schaffer, J. (1999). Older and isolated women and domestic violence project. Journal of Elder Abuse & Neglect, 11(1), 59–77. | yes | yes | no | no | yes | yes | no | yes | no | no | 5 |
| Smith, J.R. (2015) Expanding Constructions of Elder Abuse and Neglect: Older Mothers' Subjective Experiences, Journal of Elder Abuse & Neglect, 27:4–5, 328–355. | yes | yes | no | yes | yes | yes | yes | yes | no | yes | 8 |

(Continued)

Table 2. (Continued)

| Title/ author | Clear statement of research aims? | Appropriate qualitative methodology? | Description of setting and context? | Appropriate research design to address research aims? | Recruitment and sampling strategy is described and appropriate? | Data collection strategy described and justified? | Data analysis described and appropriate? | Findings supported by sufficient evidence? | Evidence of reflexivity? | Ethical issues taken into consideration? | Total score |
|---|---|---|---|---|---|---|---|---|---|---|---|
| Souto, R. Q., Merighi, M. A. B., Guruge, S., & de Jesus, M. C. P. (2015). Older Brazilian women's experience of psychological domestic violence: a social phenomenological study. International Journal for Equity in Health, 14(1), 44. | yes | yes | yes | yes | yes | yes | yes | yes | no | yes | 9 |
| Souto, R. Q., Guruge, S., Merighi, M. A. B., & de Jesus, M. C. P. (2016). Intimate partner violence among older Portuguese immigrant women in Canada. Journal of Interpersonal Violence, 34(5), 961–979 | yes | yes | yes | yes | yes | yes | yes | yes | no | yes | 9 |
| Spencer, D., Funk, L. M., Herron, R. V., Gerbrandt, E., & Dansereau, L. (2019). Fear, defensive strategies and caring for cognitively impaired family members. Journal of gerontological social work, 62(1), 67–85. | yes | yes | no | yes | yes | yes | yes | yes | no | yes | 8 |
| Teaster, P. B., Roberto, K. A., & Dugar, T. A. (2006). Intimate partner violence of rural aging women. Family Relations, 55(5), 636–648. | yes | yes | no | yes | yes | yes | yes | yes | no | yes | 8 |
| Tetterton, S., & Farnsworth, E. (2011). Older women and intimate partner violence: Effective interventions. Journal of Interpersonal Violence, 26(14), 2929–2942. | yes | yes | no | yes | no | no | no | no | yes | yes | 5 |

(Continued)

**Table 2.** (Continued)

| Title/ author | Clear statement of research aims? | Appropriate qualitative methodology? | Description of setting and context? | Appropriate research design to address research aims? | Recruitment and sampling strategy is described and appropriate? | Data collection strategy described and justified? | Data analysis described and appropriate? | Findings supported by sufficient evidence? | Evidence of reflexivity? | Ethical issues taken into consideration? | Total score |
|---|---|---|---|---|---|---|---|---|---|---|---|
| Yan, E. (2015). Elder abuse and help-seeking behavior in elderly Chinese. Journal of Interpersonal violence, 30 (15), 2683–2708. | yes | yes | yes | yes | yes | yes | yes | yes | no | yes | 9 |
| Zink, T., Regan, S., Jacobson Jr, C. J., & Pabst, S. (2003). Cohort, period, and aging effects: A qualitative study of older women's reasons for remaining in abusive relationships. Violence Against Women, 9(12), 1429–1441. **R: 347** | yes | yes | no | yes | yes | yes | yes | yes | no | no | 7 |
| Zink, T., Jacobson, C. J., Regan, S., Fisher, B., & Pabst, S. (2006). Older women's descriptions and understandings of their abusers. Violence Against Women, 12(9), 851–865. | yes | yes | no | yes | yes | yes | yes | yes | yes | yes | 9 |
| Zink, T., Jeffrey Jacobson Jr, C., Regan, S., & Pabst, S. (2004). Hidden victims: The healthcare needs and experiences of older women in abusive relationships. Journal of Women's Health, 13(8), 898–908. | yes | yes | no | yes | yes | yes | yes | yes | yes | yes | 9 |
| Zink, T., Jacobson Jr, C. J., Pabst, S., Regan, S., & Fisher, B. S. (2006). A lifetime of intimate partner violence: Coping strategies of older women. Journal of Interpersonal Violence, 21(5), 634–651. | yes | yes | no | yes | yes | yes | yes | yes | yes | no | 8 |

45–47, 49–51, 53, 54, 56, 57, 59, 61–66, 69, 70, 72–75, 77–79] lacked adequate descriptions of the study setting and context.

## Descriptions and patterns of types of violence

Older women described IPV, family violence and elder abuse of various types, perpetrated by a range of perpetrators [Table 1]. Among the specific types of violence reported in the articles in this review, across IPV, elder abuse and family violence, physical violence was most frequently reported [27, 32–54, 57, 60–63, 66, 69–80], followed by emotional/ psychological [28, 32, 36–39, 41–51, 53, 54, 56, 60–62, 66–80], economic/ financial [34–36, 39, 41, 43–45, 48–50, 61, 62, 64, 68, 71, 72, 74–78], sexual [27, 33, 34, 40, 42, 49, 50, 54, 57, 59, 60, 67, 72, 74–76, 80], verbal [32, 40, 45, 52, 62, 68–70, 73, 77], controlling behaviors [45, 48, 49, 51, 53, 64, 67, 70, 76, 77], and lastly, neglect [28, 35, 39, 61, 62, 68, 71, 76].

Older women's experience of IPV was the most frequent form of violence reported (42 articles) [27, 29, 33, 34, 36–38, 40–42, 45–56, 59–65, 67–80]. Older women described on-going instances of neglect, verbal abuse and financial exploitation in a study conducted in India [41], in other cases, physical violence characterized earlier and on-going experiences of violence within intimate partner relationships [37, 40, 47, 54]. IPV in particular was described by older women as occurring throughout different stages in the relationship, spanning their youth and into older age. Older women often experienced an escalation of IPV and controlling behaviors despite the age and/ or illness of their partner [36, 40, 46, 61, 69, 77]. Changing relationship dynamics due to ageing–including a husband's retirement, children leaving the home, women wanting to engage in activities outside of the home, or diagnosis of a chronic or terminal illness–triggered escalating IPV [36, 40, 46, 47, 56, 69]. Shifts in types of violence, from predominantly physical violence to predominant psychological abuse and neglect, were commonly described in studies that encompassed previous and on-going IPV [34, 51]. Studies focused on IPV commonly described both previous and on-going violence, and a smaller number described only or primarily violence experienced while aged 50 or above [64, 69].

Violence occurring within the family was discussed in 15 articles [35, 39, 43, 44, 46, 48, 49, 55, 63, 66, 68, 69, 71, 76, 79], with perpetrators including family members not including children [27, 43, 44, 48, 49, 59, 63, 68, 69, 71] and adult children [35, 39, 48, 66, 76, 79]. Studies captured instances of physical and verbal aggression by mentally ill adult children against older women [66], violence enacted by elderly with dementia against older women who were caregivers [46], and forms of neglect, financial exploitation and emotional abuse enacted by family members, including children [44]. The majority of these studies focused on violence experienced in older age, while one study explored dynamics of abuse between children and mothers across the lifespan [35, 39].

10 articles reported on experiences of elder abuse [28, 32, 35, 43, 44, 55, 57–59, 78] with perpetrators including community members [43, 44], caregivers [57, 59], nursing home residents [58] and health care providers [28]. Types of elder abuse included verbal abuse, physical assault and inappropriate sexual advances [58] and sexual assault [59].

Financial control spanned instances of elder abuse, family violence and IPV [43, 53, 62, 64, 76, 78], and was described as co-occurring with and resulting in other forms of violence. Financial exploitation could result in emotional and/ or physical violence if older women resisted control [62, 64]. An older woman explained that in the context of her relationship with her husband, "If I did not follow his control [over money], he would be verbally abusive" [64].

Themes and sub-themes identified through coding are displayed in Table 3.

**Table 3. Themes, sub-themes and quotations.**

| Theme | Sub-theme | Illustrative quotation[s] | Supporting references |
|---|---|---|---|
| **Intersection of ageing and violence** | | | |
| | Suffering, loneliness, regret and guilt | "I lost my whole, beautiful life. I have a lot of anger in my heart. . .Today, I am real angry about all the years, the good, beautiful years I could have had. I am angry, because I was a good, loyal wife. . . With the wisdom I have today, I would have gone out with anyone but him. Sixty wasted years" [37] | [33, 34, 37, 38, 40, 42, 44, 45, 47, 50–52, 54, 56, 62, 66, 75, 78, 79] |
| | | "There was violence along with suffering for many years; it was a suffering, but I had a goal behind all this suffering, to have my children grow, get married and get an education. . .. I don't know if the suffering was worthwhile for me, I don't know if it was worthwhile as it was very difficult. Today I look and say that I was a heroine, I was a heroine myself, with all the things I went through during the 40 years." [42] | |
| | | "When I was bringing up the children, I thought of nothing else. . . I just wanted to bring them up. Actually, my forgiveness was for the sake of the home and the children, without any consideration for myself; I did not value myself at all. I was the doormat of the entire household. . . When there was anger and quarrelling, none of the children came to ask about it. . . They never said anything. When he raised his hands, they did not go and ask him why he was hitting their mother. Nothing. As if they didn't care. I am real angry. They got used to the fact that mom gets everything done. I had big expectations of life, I gave a lot, and today I am really alone. The worst pain is from the children. Perhaps if I were stronger, I could have changed things around. But I gave up a lot. I gave in." [38] | |
| | Violence, ageing and vulnerabilities | "When I was younger, I could overcome him faster, save myself, now that I'm old and I have diabetes, now I have to be faster, and I got triglycerides in my blood. Now I'm afraid for my life, afraid he [son with schizophrenia] will kill me." [39] | [33, 37, 39, 40, 43, 44, 47, 48, 52, 53, 55–57, 59, 61, 62, 64, 66, 76] |
| | | "Because of my nerves, my blood pressure was 200/100. My sugar was skyrocketing, my cholesterol also. Since we've been living apart, everything has cooled down. When we lived together, my whole body was sick. I was hurting. I was worried that I had cancer. I couldn't believe what he had done to me. I was going to the doctor for checkups and tests all the time. I was sick with fear. My nerves made me sick. The doctor knew I had problems at home. He would say 'Ilana, you are nervous.' He gave me some pills, but nothing helped. I did not sleep. Nerves make a person sick. They make a woman sick. A sick woman without sickness." [37] | |
| | | "I reached the point where. . . I didn't care if I went and got my medicine. I would have to argue with him that I needed $12 just to go to the clinic to get my pills." [61] | |
| **Perpetrator-related factors** | | | |
| | Ageing perpetrators and continuity of abuse | "Although Mrs. V. had not been hit in many years, she was submissive to her husband and distraught about the continuing marital rape. Among the tactics used by Mr. V. to control his wife were prohibiting her from driving, working outside of the home, or managing money." [59] | [36, 41, 45, 47–51, 56, 59, 61, 72, 76, 77] |
| | | "Fifty years went by. I lost my whole life. He made me into an imbecile. . .. When I needed to buy something for the children or for myself, I had to ask him for money and he made me bring him the receipts. I have no friends, no family here; he wouldn't let the children come into the house. . . Because of the paralysis (CVA) I walk sort of crooked. He walks behind me, imitating me and calling me names like 'the limping,' 'the paralyzed'. Instead of feeling sorry about what happened to me, he laughs. 'Old whore' he called me . . . A month into my marriage, he beat me. I was pregnant. My mother was standing there and said to me: 'Be patient with him. Treat him well, take care of him and everything will be OK." [47] | |

*(Continued)*

**Table 3.** (Continued)

| Theme | Sub-theme | Illustrative quotation[s] | Supporting references |
|---|---|---|---|
| | Perpetrator's illness as a cause of violence | "We couldn't get into the house one day; the key wouldn't go in for some reason. He went berserk, kicking the door and I said, "Brian calm down, we'll go to the other door." He just kept kicking; he was just in this rage. So I backed off and went around and opened the door and came. It turned out there was damage done; he almost kicked the door in and [doctor] said in hindsight that I could have called the police then. I could have reported that because I was scared." [69] | [40, 46, 69] |
| | | "Look, I don't know what's going on with my husband, he's never been like that, never hit me before. I'm really worried about him, he's been changing so much [. . .] We have been married for 47 years. After he assaults me, he behaves as if he had done nothing, he seems another person." [46] | |
| | | "He used to work, [he was] a construction worker, and then suddenly he became agitated one day, and threw me against the wall. I cracked my head open; they stitched me up in the hospital and sent me right back home. Sometimes, he grabs me—by the stomach, by the throat, starts running, pushes me aside, and runs from room to room. I can't rest, can't watch TV, it bothers him. . .. He had become someone else, not the person I knew, so we went to the doctor." [40] | |
| **Social and gender norms regarding response to violence** | | | |
| | Silence, stigma and family | "I had a goal that my children would reach something good, and thank God, there was no other way, there was no other way. . . what I have suffered for so many years and I didn't know. I knew how to get out for the kids' sake, but not for myself. The kids get married, and go on with their lives, and I am left, left with all I have gone through. It is so difficult to speak about it, the same pain and with the same person, and today I look on my plight and I cannot leave him. . . ." [42] | [33–35, 38, 40–42, 46, 48, 57, 65–69, 71, 73–76, 79, 80] |
| | | "If I complained about him, he said that when I called the police, before the police arrive, I'd be dead. I did not know that there is help for intimate partner violence cases. I did not know because I had no friends; I did not talk to anyone! My life was from home to work and from work to home. He beat me sometimes." [81] | |
| | Perceptions of abuse and violence as normal | "I mean I suppose you could say I have been abused, I've never been badly beaten, but I have been hit and with all the temper and that sort of thing, but then there was never anywhere to go and I'm really not aware that there's anywhere specifically for older people and I'm not aware that they even do anything." [55] | [27, 29, 32, 41, 55, 70, 71, 74, 75] |
| | | "Most of the time, they [physicians] think you are just getting a little carried away, you are a little high-strung, you are very nervous, you have al- ways been this way, so calm down. . .So I didn't go to the doctor when he beat me so badly. It's a little embarrassing at my age." [74] | |
| **Lifelong IPV** | | | |
| | Continuation of patterns of IPV in old age | "He started beating me on the second day of our marriage, he's been hitting me all these years." [71] | [33, 40, 49, 59, 70–72, 77] |
| | Earlier experiences of violence | "He started beating me on the second day of our marriage, he's been hitting me all these years." [71] | [48, 49, 57, 61, 70, 71, 80] |
| | Cumulative impacts of violence | "The worst thing is that so many years of abuse caused me many health problems, especially with my nerves, and depression. This was due to mistreatment. No one can have happiness or live well dealing with so much trouble. I also have other health problems, but the worst for me are those related to my nerves, depression, and lack of sleep. I have back problems; high cholesterol, ulcers, anemia and I have a liver problem. . . a lot of problems! It never ends. Even after getting divorce, we still suffer the consequences." [81] | [34, 37, 41, 47, 50, 51, 54, 67, 68] |
| | | "I have a problem with my stomach. I did five tests and nothing was found! It is the anger I swallow. I have this pain in my stomach because the anger I feel of him." [68] | |
| | | "Bruises heal in time but words last forever. When you are told over and over how stupid, ugly, and insane you are, you really believe it. I am not financially or physically capable of going anywhere." [51] | |

*(Continued)*

**Table 3.** (Continued)

| Theme | Sub-theme | Illustrative quotation[s] | Supporting references |
|-------|-----------|---------------------------|------------------------|
| **Needs of older women affected by violence** | | | |
| | Social and community support | "If my friends knew the truth about who I was living with, then they would become really angry with me. I was losing contact with my friends because they were saying, "How could you let him treat you like this, particularly when you are in such dire need of support?" It was easier for me to just be quiet, but it's very difficult to go through such an abusive situation without having friends to talk with, though I did lose some friends." [64] | [41, 45, 52, 64, 65, 68, 75–78] |
| | | "I never invite any of my friends or relatives to come home, because of the fear that he will insult them. None comes to visit me, because I have stopped calling them" (Participant in IDI)" [41] | |
| | | "I have talked to them [my neighbours]. I have asked them to help me. The neighbours know everything, but they keep quiet! They do not want to get involved. No one comes here. No one! Only you came here today [crying]" [68] | |
| | Barriers to accessing services | "My internist really could not deal much with this [IPV]. I mean he saw my husband as a patient also. He [spouse] was a brittle diabetic, and then he had a heart condition. . . he was a sick old man." [74] | [28, 29, 52, 54, 55, 57, 74, 76] |
| | | "My family doctor is a good friend. . .didn't involve him because I didn't get. . .really hurt. I mean, I was choked, but I didn't get my eyes beat up or. . .but no I wouldn't have gone to him." [74] | |
| **Coping mechanisms** | | | |
| | Leaving a relationship | "I didn't like the way my daughter-in-law treated me. So I asked my son to find me another place to stay. Another son of mine was here, so he found a place for both of us to stay." [76] | [33, 34, 37, 38, 42, 45–48, 57, 65, 69, 70, 73, 76] |
| | Isolation, substance use and emotion-based coping strategies | 'I coped by going into my own private world; I took Valium . . . I saw myself as a failure and felt sorry for myself. [50] | [33, 34, 37–39, 47, 50, 51, 61, 68, 69, 73, 79] |
| | | "Why would I need this kind of life? How can a man do things like that? Why did I agree to that? What do I have inside me today? It is all empty; an empty shell. What am I left with? Nothing. All together, my entire life was for nothing, a big loss. . .I destroyed it all. I gave up on myself, became non-existent. I think I am a lost case. I am the loser in all this. What is left? I am all eaten up. I have no emotional strength left. I don't feel like doing anything." [47] | |
| | Behaviors to enhance safety | "The two years were coming up for the restraining order, I start getting these nightmares he's going to be at my door wanting to move in. I was living here, and he was living in [place] about a mile from home. So I go back to court and apply for renewal of the restraining order, and I am told there is no such thing as a renewal, you just apply again." [70] | [36, 67, 69, 70, 75] |

## Intersection of ageing and violence

A number of sub-themes emerged emphasizing the interconnections between the experience of ageing amongst older women, and dynamics, impacts, experiences and perceptions of violence.

**Suffering, loneliness, regret and guilt.** Older women emphasized suffering, loneliness, regret and guilt in their accounts of living and coping with violence, particularly psychological violence [34, 37, 38, 40, 42, 44, 45, 47, 50–52, 54, 56, 62, 66, 75, 78, 79]. Within the context of IPV, women described experiences of loneliness in terms of detachment from family members, including abusive partners and adult children, who often criticized older women's responses to violence [33, 34, 37, 38, 40, 42, 56]. Respondents linked regret with time and age, emphasizing previous decisions, lost opportunities, and wasting time due to living with an abusive partner [33, 34, 37, 45]. One respondent said, "I was an idiot woman. No woman lives like that, cooking and serving him after the beating. . . I say that I was an idiot" [42]. Older women expressed feelings of guilt over the abuse they experienced, and regret and guilt for exposing

their children to violence [38, 45, 50, 52, 54, 66, 79]. Several studies linked suffering, regret and loneliness specifically to psychological violence, which was described as more prominent in older age, pervasive and damaging to social relationships and self-esteem [51, 56]. The studies that explored these themes primarily encompassed accounts of violence experienced *through-out* intimate relationships–while women were younger and through to older age. These experiences were described and conceptualized by older women as interlinked and continuity of victimization by intimate partners was emphasized, rather than viewing women's experiences of violence in older age as distinct or separate.

**Violence, ageing and vulnerabilities.**   Older women described that ageing diminished their physical and emotional capabilities to cope with experiences of violence [33, 37, 39, 47]. This sub-theme appeared in 12 manuscripts [33, 37, 39, 40, 43, 47, 48, 53, 55, 59, 62, 76] and was expressed in relation to various forms of violence–IPV [33, 37, 47], including violence perpetrated by a spouse due to dementia [40], violence in the context of a new relationship or second marriage [48, 53, 55, 59, 62], violence perpetrated by a mentally ill child [39], violence perpetrated by children-in-law [76], and elder abuse [43]. These studies primarily focused on current experiences of violence of older women, as changes in physical and emotional capacity to cope was described in relation to present victimization. As a result of diminishing physical and cognitive capacities of ageing, old women experienced vulnerabilities and dependency dynamics–with partners, adult children and caregivers–that exposed them to situations of abuse [44, 47, 52, 56, 57, 61, 64, 66]. A mother of an adult son with schizophrenia explained, "When I was younger, I could overcome him faster, save myself, now that I'm old and I have diabetes, now I have to be faster. . . Now I'm afraid for my life, afraid he will kill me" [39]. Women reported that lack of financial autonomy, often compounded by years of controlling behaviors perpetrated by a violent spouse, was a central factor in women remaining in abusive spousal, caregiving and family relationships [44, 47, 52, 56, 64].

## Perpetrator-related factors

Some included studies reported on perpetrator-related factors that initiated or exacerbated forms of violence against older women.

**Ageing perpetrators and continuity of abuse.**   Older women emphasized contexts surrounding IPV in which the perpetrator continues to exercise control, power, and violence, despite their failing health and old age [41, 47–49, 51, 56, 59, 72, 76, 77]. Women also described shifting forms of violence, predominantly from physical and/ or sexual to psychological violence and controlling behaviours [36, 45, 50, 51, 61, 72]. While sometimes the experience of physical and/ or sexual violence declined, psychological violence persisted and sometimes escalated [50, 51, 72]. While describing the impacts of continual and intense psychological violence, one woman said, "he destroys you; you are not even a person anymore" [72]. Controlling behaviours were also experienced in the context of cultural norms; for example, in a study of Sri Lankan immigrant women in Canada, older women described forms of control enacted by children and children-in-law. One older women reported, "[h]e [the son-in-law] thinks that I am a widow and why should I have anything on my own name and why can't I give everything to them and just be a slave to them" [76].

**Perpetrator's illness as a cause of violence.**   This sub-theme only emerged in three manuscripts [40, 46, 69], however, it is the only instance among the included studies in which older women described first or new experiences of IPV in older age. Older women who provided care for spouses with dementia reported aggressive behavior, physical violence, and verbal abuse [40, 69]. In one study, a woman reported, "I don't know what's going on with my husband, he's never been like that, never hit me before. I'm really worried about him, he's been

changing so much [. . .] We have been married for 47 years. . .he seems another person [46]." Another study found that women who had experienced lifelong IPV understood dementia-related violence as a continuation of aggression, dominance and abuse, whereas women who had only been exposed to dementia-related violence took solace in a diagnosis, felt grief over the loss of their spouse as he used to be, and tried to maintain intimacy and affection in a previously caring and loving relationship [40].

### Social and gender norms regarding response to violence

Older women described the ways in which social and gender norms shaped their experiences of and responses to violence.

**Silence, stigma and family.** Descriptions of social and gender norms that encouraged women to stay in abusive marriages and prioritize children's needs above their own were common across studies [33–35, 38, 40–42, 46, 48, 57, 65–69, 71, 73–76, 79, 80]. Older women described several social norms that shaped their past decisions in response to violence including: silence surrounding violence and the reporting of violence [41, 80], fears of shame and stigma related to leaving a marriage [65, 69, 73], and ideals of being a good mother by putting up with violence for the sake of her children [38, 42, 74]. One woman explained, "There was violence along with suffering for many years;. . .but I had a goal behind all this suffering, to have my children grow, get married and get an education. . .. I don't know if the suffering was worthwhile for me, I don't know if it was worthwhile as it was very difficult" [42]. Remaining in a relationship as a strategy was often employed due to older women feeling obligated to care for an abusive partner who was now sick or unable to live alone [33]. One respondent explained, "If I leave him, it's not good. My conscience won't allow it. At his age, 76, it's not nice to leave and neglect him. I don't have feelings for him (because of the violence). I respect him because he's old and because he's my husband, I have to care for him" [33]. These studies primarily focused on previous and current experiences of violence; social norms predominant when women were younger shaped prior and current responses, while one study of Sri Lankan immigrant older women focused on social norms governing current decisions relating to women's responses to abuse from children and children-in-law [76].

In several cases, remaining in the relationship was a coping mechanism of last resort, given the multiple barriers present to women leaving the relationship, whether with an intimate partner, other family member or caregiver [48]. Women also described strong beliefs in social norms that supported staying with a sick or frail abusive partner or abusive child [33, 35, 40, 46, 57, 66, 76]. Many women viewed seeking help and confiding in others as embarrassing and unacceptable; one woman explained, "I was ashamed. I just didn't want to admit that's the situation I was in" [80].

**Perceptions of abuse and violence as normal.** In some of the included manuscripts, older women perceived violence as normal, sometimes explaining that they preferred not to term their experiences as abuse or violence [32, 41, 55, 70, 71, 74, 75]. Older women infrequently perceived verbal and emotional abuse as violence [32], and some women did not identify as a victim of violence [55, 71]. One woman described her process of realizing that her experiences were forms of abuse, "Well, I really didn't recognize it as abuse. And as soon as I got that message, I felt that I got on a very clear track. . ..Now, I know what I'm dealing with and I can do something about it" [74]. Moreover, service providers and the legal system, often failed to recognize financial exploitation or verbal abuse as abuse [41, 74], or that older women could be affected by IPV [75]. In rural Kentucky, USA, older women explained that the longer they were in the relationship with their abuser, the more the violence became more normalized and accepted [70]. Studies also emphasized how ageist attitudes normalizes forms of coercive

control, enabling abusers to take advantage of older women's age, frailty, and illness, for example, appropriating part or all of the victims' property [43, 44, 57].

### Lifelong IPV

Many older women described experiences of IPV throughout their life-course. Several sub-themes were identified related to lifelong patterns of violence, cumulative consequences of IPV, and linkages of violence in older age to earlier experiences of violence.

**Continuation of patterns of IPV in old age.** Older women described experiences of IPV in older age as a continuation of the patterns of violence experienced throughout the relationship [33, 40, 49, 59, 70–72, 77]. Several articles described years to decades long relationships characterized by IPV [40, 70, 71, 77]. For example, older women living in rural Kentucky, USA explained that the longer they were in the relationship with their abuser, the more the violence became more normalized and accepted [70].

**Earlier experiences of violence.** Associations between older women's earlier experiences with violence, including witnessing of violence as a child, and current experiences of IPV, were discussed in several articles [48, 49, 61, 70, 71, 80]. For example, in a study by Roberto and colleagues, many women who had experienced physical abuse as a child or young woman interpreted controlling behaviors as love, and did not recognize emotional abuse later in life until the abuse became physical or affected their health [61]. Linkages were also uncovered between experiences of abuse as a child or young woman with current abuse by their adult children [57, 71].

**Cumulative impacts of violence.** Older women described several consequences of experiences of lifelong IPV. In one study, older women related the impacts of lifelong violence to that of a chronic illness, which alters or limits one's quality of life [47]. Older women frequently linked experiences of violence with physical health consequences, including bodily pain, reduced mobility, and hearing problems, [37, 47, 54, 67], as well as mental health and emotional impacts, including depression [41, 50, 51, 54, 67], anxiety [54, 67], panic attacks [54], suicidal ideation [41], loneliness [34, 51], and loss of self-esteem [34, 50, 51, 54].

### Needs of older women affected by violence

Older women who reported exposure to violence described various needs in terms of social support, access to services, and issues accessing these services due to their age.

**Social and community support.** Older women commonly described isolation from family and friends, and a lack of social and community support as a result of violent and controlling behaviors from an intimate partner [41, 45, 52, 64, 65, 75–78]. One older woman stated: "I cannot remember, not one time, not having the hell beat out of me. Black and blue, I wasn't even allowed outside. I couldn't open my mouth, I couldn't talk, I couldn't have friends. I had neighbors, and they didn't know me . . .He threatened to kill me if I ever told anyone what was going on" [70]. Additionally, factors that were reported to impede access to social and community support included being an immigrant with limited language skills [67, 68, 76], and living in rural areas with strong norms against reporting IPV [52].

**Barriers to accessing services.** Several articles identified specific barriers for older women to access services and for health care utilization, including lack of awareness of services [52, 54, 55, 57, 74, 76]. Older women reported several concerns when interacting with health care providers, including health care providers' assumptions that older women could not be experiencing violence due to their age, minimization of forms of abuse common to older women, and lack of confidentiality when using the same provider as their spouse [74]. One respondent

explained, "And when you go to the doctor. . .they run down the list. . .and then it's always, you know, "Well, is it abuse?" "Well, yes emotional." "Well, what kind of emotional?" "Verbal." "Oh, OK." And they mark it, and that's it" [74].

## Coping mechanisms

Older women reported various approaches to coping with the experience and impacts of different forms of violence, often employing several different coping mechanisms such as leaving relationship with an abuser and emotion-based coping strategies such as alcohol or drug usage, in order to navigate difficult decisions, maintain their health and well-being, and protect other family members in the context of exposure to violence.

**Leaving a relationship.** In 11 of the included manuscripts older women described remaining in an abusive relationship, family context or caregiver relationship, as a form of coping [33, 34, 38, 42, 45, 46, 48, 57, 65, 70, 73], and in six manuscripts, leaving a relationship was employed as a coping mechanism [42, 45, 61, 69, 76, 77]. In one study, older women explained that they had previously not been able to leave a relationship with an intimate partner for the sake of their children, whereas once their children had left the house, they felt freer to reject violent behavior [42]. Older women's own health problems were described as a trigger for choosing to leave an abusive relationship [61].

**Isolation, substance use and emotion-based coping strategies.** Older women described isolating themselves from family, friends and social support, using alcohol or drugs to cope with experiences of violence, and reframing experiences of violence, often through minimizing experiences [33, 34, 37–39, 47, 50, 51, 61, 68, 69, 73, 79]. Older women explained that if they were to seek support, family or friends would blame them for their experiences of violence, leading women to choose social isolation as a coping strategy [50, 69]. Older women also described using drugs and alcohol as a means to numb themselves to their daily experiences of violence [50]. One woman explained, "He (my husband) got his medical partner to prescribe Valium for me in the 1970's and I am still taking it, especially when I feel hopeless and in despair. I know that I am addicted to it and worry that at 68 years I will never be able to survive without them." [50]. Older women also reported employing forgiveness of violent and controlling intimate partners as a coping mechanism [34, 38]. Older women who remained in a relationship with their abuser often described employing emotional detachment as another coping strategy [33, 37, 47, 69]. Lastly, older women described how they reframed their experiences of abuse, by excusing abusive spouses for their actions or employing strategies to deliberately diminish the severity of abuse, such as forgetting experiences of abuse [34, 38, 61, 68, 73]. While emotional detachment was described as causing isolation and loneliness, older women also perceived it as a form of "inner resistance" [37], a vital means of opposing intimacy and connection with an abusive partner, and as particularly vital in the case of IPV, where the safety of a woman's home is threatened by violence [47].

**Behaviors to enhance safety.** Older women described taking actions in order to enhance their own safety in the face of violence [36, 67, 69, 70, 75]. In some instances, older women first called police or applied for formal legal support, such as a protection order, in the face of violence. In one study, a woman explained, "I called the police because he [my partner] pushed me down on the countertop and poured a cup of tea over me. It was as though he wanted to strangle me. They took him into custody for 24 hours" [36]. In several instances, legal authorities, including police, provided limited support, leaving women unable to find long-term solutions to the violence they experienced [70].

## Discussion

This systematic review was motivated by a need to improve understanding of similarities and differences in dynamics, patterns and experiences of violence against older women, in a context whereby the vast majority of research, evidence, policy and service provision is targeted towards women of reproductive age. We reviewed available qualitative studies on violence against older women in order to address existing gaps in evidence and data. We also sought to provide insight into the lived experiences of older women experiencing violence, and an understanding of the types and patterns of violence, perpetrators of violence, and health impacts of violence among older women. The included studies primarily address IPV, with fewer emerging from the older adult mistreatment framework. Most research examined specific types of violence in isolation, for example, IPV or abuse from an adult child, and there were no examples of studies that included polyvictimization or experiences of any type of violence against older women. The strong emphasis on older women's experiences of IPV gives voice to the experiences of older women subjected to violence and shows how it can persist over time; however, some sites, perpetrators and types of violence against older women may be excluded from view, including that of violence enacted by other family members and non-family caregivers and of women living in institutional care.

The findings in our review confirm results from prior reviews, systematic and otherwise, of similar bodies of literature. For example, Pathak et al.'s review of IPV against older women noted a decline in physical violence against older women, whereas other forms of violence remained stable or increased, a finding that was reflected in our data [23]. Some of the studies included in the present review also confirm partners' retirement and children leaving home as precipitating factors for increase of IPV against older women, indicating points for potential intervention and support for older women. In a review of qualitative literature on IPV against older women, Finfgeld-Connett noted that older women actively choose coping strategies that enable them to "make the best of their situations" [20], a conclusion that is also supported by some of the results of our review. In other cases, staying in a relationship with an abuser appears to be driven by gender norms and feelings of duty towards a partner. In addition [34, 38], coping strategies such as use of alcohol and other harmful substances appeared to result in poor health and lack of well-being [34, 37–39, 47, 50, 51, 61, 68, 69, 73, 79]. Recurring themes emphasizing the pervasive impact of violence against older women on physical and mental health, relationships, social networks, hope and sense of well-being, in our systematic review and other previous reviews, indicate the importance of taking violence against older women, in all its manifestations, seriously as a public health and human rights issue. As was identified in previous reviews, there is relatively little evidence concerning the emergence of violence in later life, particularly in the case of IPV. In the case of the majority of studies included in our review, older women described shifting but continuous patterns of violence throughout the life-course, although a small sample of studies identified new relationships and dementia of an intimate partner as factors precipitating the violence [40, 46, 48, 53, 55, 59, 62, 69].

Comparing the IPV-specific evidence generated in this review to the existing evidence-base on IPV against women of reproductive age, some notable continuities and differences are evident. Firstly, our findings confirm the extensive impact of IPV exposure on physical and mental health, which has been widely researched amongst women of reproductive age [2, 82–85]. However, our data indicate that IPV amongst older women is commonly experienced in the context of exposure to lifelong IPV, and that the physical and mental health impacts are cumulative, compounded by ageing processes, and often exacerbated by changes in social situation also triggered by ageing. Ability to employ physical or cognitive coping mechanisms that had been effective earlier in life may diminish for older women [33, 37, 39, 47]. In addition,

alongside depression, anxiety and post-traumatic stress disorder, which are the most commonly measured and reported mental health impacts of IPV amongst women of reproductive age [86–88], older women discussed hopelessness and regret as pervasive and important psychosocial impacts of IPV in older age. There may be some similarities between younger women's experiences of shame and stigma [89–93] and older women's feelings of regret, however, regret and hopelessness may be specifically central to older women's experiences of violence, particularly IPV. Secondly, our results confirm that exposure to IPV is often linked to experiences of violence in childhood; older women in studies included in this review indicated that growing up in families where violence was commonly witnessed and experienced was interlinked with exposure to IPV in adulthood and through to older age, a finding that is evident in data on women of reproductive-age [94–96]. Thirdly, there appear to be common challenges for women of reproductive age and older women in leaving an abusive relationship, including perceptions of the importance of remaining in a relationship for the sake of children, indicating the commonality of the importance of social and gender norms in driving decision-making [97–101]. Implications garnered from research with women of reproductive age experiencing IPV are relevant here; similarly, it should not be assumed that older women want to or can leave an abusive situation, and services provided should recognize and be sensitive to this. Finally, our findings highlight specific issues for consideration in the case of violence against older women, including changes in type and prevalence of controlling behaviours [36, 45, 50, 51, 61, 72, 77] and forms of financial control that occur alongside IPV [43, 53, 62, 64, 78]. These behaviours have the potential to significantly restrict options and limit ability for older women experiencing violence, even more than in younger women. Currently however, these may be under-recognized as specific risk factors for older women.

Global research on violence against women has increasingly explored the significant influence of social and gender norms on prevalence of and risk factors for violence against women of reproductive age [102–105]. Our findings indicate that social and gender norms also continue to strong influence older women's responses to and experiences of violence. Older women described social and gender norms as shaping their decisions to stay in relationships, to provide care for an abusive spouse, and often as reinforcing shame and social isolation. There is substantial overlap between norms identified in this review with the existing evidence-base on social and gender norms on women of reproductive age, for example, the norm of keeping violence victimization private and overall injunctions concerning silence surrounding IPV. Some evidence indicates positive impacts of violence prevention interventions focused on changing social and gender norms [106]. However, these programs have not been specifically tested for feasibility and acceptability with older adults, and careful consideration of how and if addressing social and gender norms amongst older adults could result in reduced violence perpetration is needed.

Our review identified significant gaps in the evidence-base concerning older women's experiences of violence in low and middle-income countries (42 articles in HIC vs. 10 articles LMIC). Within studies conducted in high-income countries, with a few exceptions [52, 61, 70, 76, 80], the focus of the included studies was on older women from Western cultural backgrounds. The sparse coverage of several regions globally, and low and middle-income populations overall, indicates that our findings cannot be generalized to older women globally, and that there are likely important influences on and impacts of violence against older women that are currently missing from view. While we can assume that older women in low and middle-income contexts also experience violence, the existing evidence base, for both qualitative and quantitative data, fails to adequately shed light on patterns and prevalence [16, 17]. In addition, as found in our quality assessment, included articles contained very little detail on the contexts in which the research was conducted [32, 34–42, 45–47, 49–51, 53, 54, 56, 57, 59, 61–66, 69, 70,

72–75, 77–79], which makes it difficult to link the evidence from this review to specific contextual factors. Further exploration of context-specific issues such as living conditions and associated norms, for example, norms governing that older widows live with children and children's families is needed. In addition, exploration of perceptions of capabilities and appropriate social roles for older women in different socio-cultural contexts is warranted. Perceptions and experiences of ageing processes, and specific issues such as widowhood, differ significantly in different cultural contexts, and existing qualitative and quantitative data do not include these diverse factors or account for their relationship with violence against older women.

Our findings indicate that older women affected by violence need social and community support to help them cope and address the anxiety and stress associated with threats to their safety. Older women affected by violence may be particularly isolated, with social isolation concomitant with ageing compounded by social isolation due to violence victimization. Some of the studies indicated that older women do not understand or define their experiences as abuse or violence, but do seek support regardless. As such, there may be potential for services and interventions designed to address social isolation and targeted for all older women to address violence against older women. Various interventions that have been found to be effective in reducing social isolation and improving social outcomes for older persons, such as group support through discussion groups, individual support through home visiting, and psychosocial education programs, could be effective in improving social support for older women affected by violence [107]. Currently, services for older persons are overall extremely limited in low and middle-income contexts, and dynamics of social and community support for older persons vary considerably in non-Western cultural contexts. The current qualitative evidence-base does not indicate if older women affected by violence in low and middle-income contexts would benefit from similar interventions or if integrating response to and support for violence against older women into aged-care services are a feasible way to reach older women affected by violence.

In the limited number of included studies that addressed older women's experiences with and expectations of health-care providers, concerns were raised including lack of confidentiality and health care providers not taking women's abuse seriously [74]. Health care providers are in a unique position to provide support and response for women who have been affected by violence. The World Health Organization's Clinical and Policy Guidelines and Clinical handbook provide guidance for health care providers in providing woman-centred care, compassionate first-line psychosocial support, and linkages to multi-sectoral services [108]. In the case of older women, women may come in contact with primary, secondary or tertiary health care services for reasons related to chronic disease and ageing-related injuries, for example, or as care-givers for spouses or children. There is a need to explore how and where violence prevention and response for older women in the health system could be feasible and acceptable. For example, gerontologists and other specialists providing elder-care specific services could be provided tools and skills to identify and support women who may be subjected to violence. In addition, as identified in this and other reviews of violence against older women, there are factors that may act as precipitating factors for increase or initiation of violence, including changes in caregiving dynamics or retirement of a spouse, and these could be points of potential intervention and additional support for older women, especially if there is a history of past violence.

## Limitations and strengths

Several strengths and limitations should be considered while interpreting the findings of this systematic review. In contrast to previous systematic reviews, we included all qualitative

evidence concerning violence against older women, regardless of type of violence and perpetrator, allowing insights into the overall focus of the evidence-base, which revealed limited engagement with elder abuse against women and family violence perpetrated by non-partners, for example, children. Additionally, we followed a rigorous protocol, adhering to a preregistration protocol in line with ENTREQ guidelines [31]. We carried out an extensive systematic review across 11 databases, supplemented by hand searched references lists and article recommendations from 49 experts on violence against women or older adults, and therefore it is unlikely that published articles would have been overlooked in this review. We reviewed all articles in any language, apart from Farsi.

In order to minimize selection bias or for relevant articles to be missed, two authors screened all titles and abstracts and all articles at the full text review stage. At the data extraction phase, only MEL extracted relevant data, introducing the possibility of transcription errors. Despite this limitation, all extracted data was double checked by SRM to minimize potential of missing descriptive data, and both completed independent quality appraisals to minimize potential for biased assessments. Additionally, during the analysis phase, both authors coded article main findings and key quotes, and developed descriptive and analytical themes to strengthen the interpretation and synthesis of findings.

Another limitation of the findings of this review is the concentration of studies in higher-income contexts, which greatly limits the transferability of findings to low- and middle-income populations. In addition, the small number of studies conducted in low and middle-income countries entailed that comparison of patterns between high-income and low and middle-income contexts was not possible. This review was also limited by the quality of included articles. Many articles did not clearly report on study setting and context, sampling procedures, data analysis, reflexivity, and research ethics. Moreover, many articles included samples of older women across wide age ranges (e.g. 65–85 years old). The available evidence does not disaggregate findings to enable understanding of whether or how women in different age groups experience violence differently, despite significant variation in living conditions, employment and health status of women aged 50–64 vs. 65 and up, for example. This lack of specificity limited our ability to understand the differential causes, experiences and impacts of violence among specific age groups of older women.

## Implications for future research

In light of the findings from this review, as noted above, there is an urgent need to address the scarcity of research on violence against older women in low and middle-income contexts, and to expand research in high-income contexts to diverse populations and age groups who may have different risk profiles for violence in older age.

Our results indicate that the focus of the existing qualitative evidence-base is primarily on IPV in older age. It is unclear whether this research focus reflects the actual burden of IPV compared to other forms of violence against older women, and if the evidence-base currently adequately includes accounts of types of violence and perpetrators that are most significant for older women. As noted, assessment of polyvicimization in the evidence-base is lacking. As such, further studies of violence against older women that are inclusive of any type of violence, by any perpetrator, or take an open-ended approach to older women's accounts of violence, are needed. In the quantitative evidence-base, systematic reviews have focused on elder abuse and on IPV. A review of quantitative evidence on IPV identified 19 studies [15] and the review of elder abuse against women included 50 studies; as such, the quantitative evidence-base appears to capture more in terms of forms of violence against older women.

Our findings indicate that for older women who had experienced violence throughout the life-course, aspects of ageing, such as frailty, injuries, chronic disease, and cognitive decline, make coping with different forms of violence more difficult than earlier in life. Qualitative and quantitative research does not currently shed light on associations between types of violence, chronicity of violence, and physical and mental health outcomes for older women, and additional research in this area is warranted. Other themes that emerged in our review call for further research. Regret and hopelessness were commonly described as significant issues for older women; these factors appear to significantly influence well-being, psychosocial health and physical and mental health. However, these outcomes are rarely measured, and these may further impact other specific mental health and psychosocial issues for older women subjected to violence. Further research could explore if and how regret and hopelessness amongst older women differs from shame and stigma as currently measured and reported amongst women of reproductive age, and further elucidate its impacts on psychosocial well-being. In addition, economic and financial abuse appeared to be correlated and interlinked with older women's experiences of violence, and barriers to leaving abusive relationships; terminology and definitions of these forms of abuse are varied and often unclear, and measures often cover several constructs [109]. While there is some growing consistency in how economic and financial aspects of abuse are conceptualized and measured, there is more work needed on how to assess economic or financial abuse, and understand its linkages with physical and mental health outcomes.

## Conclusion

The current qualitative data available on violence against older women has important limitations, including that it is predominantly derived from high-income countries, often does not address context, is focused on IPV to the exclusion of other types of violence and perpetrators, and does not disaggregate by age group. However, our findings highlight some important issues. IPV persists into older age, and shares characteristics and impacts as in younger age groups. In some cases, there may be factors, such as a partner's retirement or illness such as dementia, which can precipitate or increase violence. As shown in quantitative reviews, physical violence tends to decrease with age while psychological abuse and controlling behaviours increase, and financial and economic abuse are important elements of older women's experiences of violence and control. Older women described being strongly influenced by social norms that dictate a sense of duty to stay in a relationship with an abusive partner, a desire to protect children, and shame and silence surrounding experiences of violence. More research is needed, particularly from LMICs to fill in the many gaps in the evidence-base. However, it is clear that action to support older women in abusive relationships is needed. Services for older people need to be aware of the prevalence and forms of violence against older women and know when to identify and respond in a sensitive and non-judgmental way, to improve prevention of and response to violence against older women.

## Supporting information

**S1 File. PubMed search strategy.**
(DOCX)

**S2 File. ENTREQ checklist.**
(DOCX)

**S3 File. PRISMA checklist.**
(DOC)

## Author Contributions

**Conceptualization:** Sarah R. Meyer, Claudia García-Moreno.

**Data curation:** Sarah R. Meyer, Molly E. Lasater.

**Formal analysis:** Sarah R. Meyer, Molly E. Lasater, Claudia García-Moreno.

**Funding acquisition:** Claudia García-Moreno.

**Investigation:** Sarah R. Meyer, Molly E. Lasater.

**Methodology:** Sarah R. Meyer, Claudia García-Moreno.

**Resources:** Claudia García-Moreno.

**Supervision:** Claudia García-Moreno.

**Writing – original draft:** Sarah R. Meyer.

**Writing – review & editing:** Molly E. Lasater, Claudia García-Moreno.

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
