## [Decision Letter · Decision Letter 0]

23 Jul 2020

PONE-D-20-10316

Violence against older women: a systematic review of qualitative literature

PLOS ONE

Dear Dr. Meyer,

Thank you for submitting your manuscript to PLOS ONE. After careful consideration, we feel that it has merit but does not fully meet PLOS ONE’s publication criteria as it currently stands. Therefore, we invite you to submit a revised version of the manuscript that addresses the points raised during the review process.

**The Reviewers considered the manuscript very positively. They also provided several suggestions to improve the quality of the study and make it suitable for publication.**

We look forward to receiving your revised manuscript.

Kind regards,

Stefano Federici, Ph.D.

Academic Editor

PLOS ONE

Additional Editor Comments:

The Reviewers considered the manuscript very positively. They also provided several suggestions to improve the quality of the study and make it suitable for publication.

Journal Requirements:

2. Please cite your published protocol article: https://www.ncbi.nlm.nih.gov/pmc/articles/PMC6550033/

3. Please include your tables as part of your main manuscript and remove the individual files. Please note that supplementary tables (should remain/ be uploaded) as separate "supporting information" files.

5. Please include a caption for figure 1.

<h1>** **</h1>

Reviewers' comments:

Reviewer's Responses to Questions

**Comments to the Author**

1. Is the manuscript technically sound, and do the data support the conclusions?

Reviewer #1: Yes

Reviewer #2: Yes

Reviewer #3: Yes

2. Has the statistical analysis been performed appropriately and rigorously? 

Reviewer #1: N/A

Reviewer #2: Yes

Reviewer #3: Yes

3. Have the authors made all data underlying the findings in their manuscript fully available?

Reviewer #1: Yes

Reviewer #2: Yes

Reviewer #3: Yes

4. Is the manuscript presented in an intelligible fashion and written in standard English?

Reviewer #1: Yes

Reviewer #2: Yes

Reviewer #3: Yes

5. Review Comments to the Author

Reviewer #1: Recommendation: Publish with minor corrections

1. Summary of Research and Overall Impression

This is an excellent summary of qualitative studies on neglect, abuse, and violence against older women. It makes an important contribution to the literature on this often overlooked population in the domestic violence field. It is well-written and well-researched.

Methodology and selection criteria for studies included in the review are clearly presented. Authors of selected studies are known for their expertise in this area of international domestic violence research on older women. The systematic review presented here complements findings from quantitative including prevalence studies.

This is a particularly appropriate report for UN Women, which has not always been open to considering older women’s experiences with domestic violence as relevant to the field of international domestic violence. The author frames the analysis in a particularly insightful way using a feminist perspective. In doing so, she effectively challenges the ageism inherent in views of older women and domestic violence as “elder abuse” that is disconnected from gender, community, and the life course.

The author chose qualitative research studies that incorporate the words of older women “in their own voices” and use a life course perspective. This very much reflects a feminist perspective. She also makes efforts to include voices of older women from developing countries, which she notes is difficult. Organizations like HelpAge International have done studies on older women and abuse from developing countries, but these studies tend not to reflect the rigor of qualitative studies undertaken by the academy based on her stated selection criteria.

2. Minor issues

101 – Instead of “older adult mistreatment” framework, the author may want to consider substituting “vulnerable older adult” framework and distinguish this from the “intimate partner violence (IPV)” framework and the “active ageing” framework that can incorporate feminist gerontology, although can also reflect a gender neutral perspective. Older adult mistreatment is a generic term, while IPV is more commonly used in domestic violence discussions involving women survivors (and the author makes a point of selecting studies that view older women as women, not elders”). The vulnerable older adult framework reflects a disconnect between women domestic violence survivors and older women, and which this reviewer would argue incorporates an ageist and gender neutral framing of older women survivors. The underlying assumption of this frame is that older victims are frail and dependent by definition: this both narrows the population to be included in the study to a subset of impaired older adults, or alternatively assumes that older women are by definition impaired, care dependent and “vulnerable”. The author has selected studies for the review that quite rightly challenge this assumption.

738 – Under limitations, the author notes that there is a dearth of qualitative studies on older women survivors of domestic violence from developing (low income) countries. This reviewer noted earlier that in fact there may be other studies (for example, those by HelpAge International – Bridget Sleap) but not reflecting the academic rigor sought in this review. However, there are studies conducted in high income countries of older women survivors of domestic violence who are immigrants from low income countries. One example that comes to mind is Guruge et al. (2010), Older women speak about abuse and neglect in the post-immigration context, conducted in Canada with Sri Lankan immigrants.

Polyvictimization is a fairly new concept in older adult abuse, and one that is not prominent in the domestic violence literature to date but is more so in the field of child abuse. Studies by Pamela Teaster and Holly Ramsey-Klawsnik, for example, have found that multiple forms of abuse/multiple abusers experienced by domestic violence victims can lead to increased trauma. If the qualitative studies did not specifically ask about polyvictimization, they may not have captured this.

Overall, these are minor issues. The charts included in the manuscript are very helpful in providing a flavor of the felt experience of older women survivors of domestic violence.

3. Other Points

Overall, an excellent review and one that can serve to educate UN Women staffers, primarily young women, about older women’s lived experience of domestic violence. This will hopefully result in their viewing older women as part of the continuum of “Girls and Women of all Ages” and not “Other”.

Reviewer #2: This is a fascinating study, drawing attention to urgent issues in this field of research and responds to a research gap that it clearly identifies at the outset. It has a well-articulated methodology, discussion and makes powerful conclusions. Further details of the analytical approach would be welcomed, however, as it is currently unclear how themes and sub-themes were reached. There is some overlap between some sub-themes, and in some sub-themes there is a lack of depth where the results could be explored further. Either further details on how the analysis was conducted would address this, or some reorganisation of themes and sub-themes to a smaller number that would allow for exploration of the results in more detail.

Further explanation of why one article in Farsi was not translated and included in the study would also be welcomed.

Reviewer #3: Thank you for giving me the chance to review this important manuscript that addresses a clear gap in the literature. It is very clear that you have done an amazing job in thoroughly going through a vat amount of literature and put a lot of attention in capturing necessary detail. Congratulations on it!

I have some general and specific comments.

General comments:

• The title of the paper is violence against older women and throughout the text you refer to older women. The definition of older women in the text is women aged 50 or older. Is this aligned with existing definition of older or is it a consequence that many other studies have focused on women of reproductive age? Should you not throughout the title and text to simply name them women aged 50 or older or is older women the correct term?

• The review stats that it is “exploring violence against women aged 50 and above, identifying types and patterns of violence, perpetrators of violence, and impacts of violence on various health outcomes for older women”. Reading the review, it seems that much more has been investigated than that, namely associate factors and consequences beyond health outcomes, such as loneliness and social isolation. This could be stated more clearly in the introduction, as I kind of expected it but did not find it reflected in the description of the reviews scope.

• Throughout the results section it was often unclear whether you are summarising violence described actual violence that older women are experiencing now or whether they refer to any violence as it sounds like in the section on loneliness (not only an issue there)? It is important to make that very clear whether women refer to past violence, potentially 20 years ago or current violence. For example, but not only there, in the section “Silence, stigma and family” it is unclear if the women refer to the current violence they experienced or past violence, potentially 20 years ago. The whole paragraph seems to refer to varies time points in which the violence happened and this needs to be clarified, especially since there is a specific sub-section for it

• Have you actually found any differing evidence by different age categories among women aged 50 or above? In the limitation section you mention that the age rages varied widely, but it would be could to situate the results into this as 50 to 64 is quite a different age category than 65 to 99, when women are also more likely to be retired.

• I am a bit surprised by the structuring of the results section and consider reworking it as it jumps from the overview of forms of violence and perpetrators in a summary paragraph to causes, consequences, risk factors to financial abuse- a form of violence experienced, norms, needs, early childhood violence as a risk factor later. While the overall heading make sense, the sub heading sometimes seem to belong somewhere else and do not flow.

• Did you actually find different results by LMIC and HIC?

• The section “Descriptions of types of violence and perpetrators” is actually a quite crucial one, but it is currently very descriptive in terms of numbers. Given that forms of violence and perpetrators are such a key objective, could you expand this section and show what kind of violence was perpetrated by the different perpetrator types or forms of abuse and whether there were any age trends among the older women.

Small, specific comments:

• Abstract p 8, line 31: grammatical mistake

Introduction:

• Page 4, line 77 needs a reference

• Page 6, line 116: Can you state if these systematic reviews said anything about the perpetrators of this violence? Which age ranges did the systematic reviews investigate?

• Page 6, line 128 grammatical error

Method:

• Did you use any time limit for the search?

• Did forms/types of violence cluster in certain countries?

Results:

• On page 14 under the heading “Descriptions of types of violence and perpetrators” you first describe the terms IPV, family violence and elder abuse, but these terms have not been described before in terms of what they mean and how they are different from each other

• On page 15 when you talk about causes of violence, which types of violence were referred to?

• Is the financial abuse section actually referring to tis as a form of violence or a cause for other violence or a co-occurrence of numerous forms of violence?

• In the discussion section you refer to health care providers response and women’s concerns regarding confidentiality, however, this was not brought up in the results at all and should have been mentioned there too.

• Implication for future research – the scarcity of research only relates to qualitative work or more generally? I understand that your review mainly found studies on IPV, but what did the quantitative reviews find? Would they support your claim?

6. PLOS authors have the option to publish the peer review history of their article (what does this mean?). If published, this will include your full peer review and any attached files.

Reviewer #1: No

Reviewer #2: No

Reviewer #3: **Yes: **Heidi Stöckl

---

## [Author Response · Author response to Decision Letter 0]

7 Aug 2020

Authors’ response to reviewers

Manuscript title: Violence against older women: a systematic review of qualitative literature

To the Editors, PLoS One

Thank you for the recognition of the contribution of our manuscript, “Violence against older women: a systematic review of qualitative literature.” In response to the reviewers’ comments, some changes have been made to the manuscript. We appreciate the reviewer’s positive comments and feel that our responses to these helpful suggestions helped improve this manuscript. The reviewers’ comments, as well as journal requirements listed, are addressed point-by-point in turn below.

Reviewer 1:

1. 101 – Instead of “older adult mistreatment” framework, the author may want to consider substituting “vulnerable older adult” framework and distinguish this from the “intimate partner violence (IPV)” framework and the “active ageing” framework that can incorporate feminist gerontology, although can also reflect a gender neutral perspective. Older adult mistreatment is a generic term, while IPV is more commonly used in domestic violence discussions involving women survivors (and the author makes a point of selecting studies that view older women as women, not elders”). The vulnerable older adult framework reflects a disconnect between women domestic violence survivors and older women, and which this reviewer would argue incorporates an ageist and gender neutral framing of older women survivors. The underlying assumption of this frame is that older victims are frail and dependent by definition: this both narrows the population to be included in the study to a subset of impaired older adults, or alternatively assumes that older women are by definition impaired, care dependent and “vulnerable”. The author has selected studies for the review that quite rightly challenge this assumption.

We agree with this reviewer that there are other ways to capture and describe the dominant conceptual frameworks in this literature. We feel that the three frameworks that we describe – older adult mistreatment, older adult protection, and IPV – are one way to capture the approaches on the literature. Rather than substituting a different framework, we have added some comments reflecting the problematic aspects of each of these approaches, including the disconnects and underlying assumptions that this reviewer rightly points out. The section now reads:

“The older adult mistreatment framework conceptualizes violence against older women as a form of elder abuse, focusing on age as the primary factor influencing vulnerability to exposure to violence. The older adult protection framework specifically understands violence within the context of care-giving and institutional arrangements, where older adults’ often be gender neutral, and the adult protection framework can result in a framing of older adults as inherently impaired and vulnerable. In addition, the IPV framework primarily understands vulnerability to violence in terms of gender inequality and partnership dynamics, which may neglect analysis of how ageing and partner violence intersect.”

2. 738 – Under limitations, the author notes that there is a dearth of qualitative studies on older women survivors of domestic violence from developing (low income) countries. This reviewer noted earlier that in fact there may be other studies (for example, those by HelpAge International – Bridget Sleap) but not reflecting the academic rigor sought in this review. However, there are studies conducted in high income countries of older women survivors of domestic violence who are immigrants from low income countries. One example that comes to mind is Guruge et al. (2010), Older women speak about abuse and neglect in the post-immigration context, conducted in Canada with Sri Lankan immigrants.

We appreciate the reviewer pointing out this relevant article, and we have included it in our review.

3. Polyvictimization is a fairly new concept in older adult abuse, and one that is not prominent in the domestic violence literature to date but is more so in the field of child abuse. Studies by Pamela Teaster and Holly Ramsey-Klawsnik, for example, have found that multiple forms of abuse/multiple abusers experienced by domestic violence victims can lead to increased trauma. If the qualitative studies did not specifically ask about polyvictimization, they may not have captured this.

We agree with this comment, and have included the following text in the Discussion section to indicate that polyvictimization is not adequately addressed in this literature: 

“The included studies primarily address IPV, with fewer emerging from the older adult mistreatment framework. Most research examined specific types of violence in isolation, for example, IPV or abuse from an adult child, and there were no examples of studies that included polyvictimization or experiences of any type of violence against older women. The strong emphasis on older women’s experiences of IPV gives voice to the experiences of older women subjected to violence and shows how it can persist over time; however, some sites, perpetrators and types of violence against older women may be excluded from view, including that of violence enacted by other family members and non-family caregivers and of women living in institutional care.”

AND

“As noted, assessment of polyvicimization in the evidence-base is lacking.”

Reviewer 2:

1. Further details of the analytical approach would be welcomed, however, as it is currently unclear how themes and sub-themes were reached. There is some overlap between some sub-themes, and in some sub-themes there is a lack of depth where the results could be explored further. Either further details on how the analysis was conducted would address this, or some reorganisation of themes and sub-themes to a smaller number that would allow for exploration of the results in more detail.

We agree that further discussion of the analytical approach is warranted, and have added the following description in the Methods section:

“Two of the authors (SRM and MEL) coded the main findings extracted from each study. We used line-by-line coding on a sub-set of articles, developing a set of over-arching themes and sub-themes for a draft codebook. The coding proceeded as an iterative process, with the two authors each separately coding the main findings using the draft codebook, discussing coding results, and refining the codebook based on overlap and redundancies identified. After all data was coded and we tallied all occurrences of codes, we further explored areas of overlap and merged sub-themes with low numbers of codes, finalizing the broad themes and focused sub-themes, displayed in Table 3.”

Further, in response to this comment and Reviewer 3’s comments regarding the Results section, we have reorganized themes and sub-themes to reduce overlap and provide more detail. The themes and sub-themes are now:

• Descriptions and patterns of types of violence

• Intersection of ageing and violence

o Suffering, loneliness, regret and guilt.

o Violence, ageing and vulnerabilities.

• Perpetrator-related factors

o Ageing perpetrators and continuity of abuse.

o Perpetrator’s illness as a cause of violence.

• Social and gender norms regarding response to violence

o Silence, stigma and family.

o Perceptions of abuse and violence as normal.

• Lifelong IPV

o Continuation of patterns of IPV in old age.

o Earlier experiences of violence.

o Cumulative impacts of violence.

• Needs of older women affected by violence

o Social and community support.

o Barriers to accessing services.

• Coping mechanisms

o Leaving a relationship.

o Isolation, substance use and emotion-based coping strategies.

o Behaviors to enhance safety.

2. Further explanation of why one article in Farsi was not translated and included in the study would also be welcomed.

We have added the following sentence to explain this: “One non-English article (in Farsi) was not reviewed as the research team could not engage a Farsi speaker to review the article.”

Reviewer 3:

1. The title of the paper is violence against older women and throughout the text you refer to older women. The definition of older women in the text is women aged 50 or older. Is this aligned with existing definition of older or is it a consequence that many other studies have focused on women of reproductive age? Should you not throughout the title and text to simply name them women aged 50 or older or is older women the correct term?

We have focused this review on women aged 50 and above as a consequence that many other studies have focused on women of reproductive age. Definitions of old and older women vary across organizations and research. However, we found that using the phrase ‘women aged 50 and older’ throughout the manuscript, while more accurate, was difficult to understand and disrupted flow of the narrative. Therefore, we added a sentence in the introduction to explain this:

“While there is no universal agreed-upon definition of older women, for the purposes of this review, we define older women as women aged 50 and above, while recognizing that aging and age are social phenomenon, and definitions vary across organizations, cultures and communities.”

2. The review stats that it is “exploring violence against women aged 50 and above, identifying types and patterns of violence, perpetrators of violence, and impacts of violence on various health outcomes for older women”. Reading the review, it seems that much more has been investigated than that, namely associate factors and consequences beyond health outcomes, such as loneliness and social isolation. This could be stated more clearly in the introduction, as I kind of expected it but did not find it reflected in the description of the reviews scope.

We have altered this statement to include the wider range of outcomes that are considered in this review. The sentence now reads:

“We aimed to identify, evaluate and synthesize qualitative studies from all countries, exploring violence against women aged 50 and above, identifying types and patterns of violence, perpetrators of violence, and impacts of violence on various outcomes for older women, including physical and mental health and social support, and women’s responses to experiences of violence.”

3. Throughout the results section it was often unclear whether you are summarising violence described actual violence that older women are experiencing now or whether they refer to any violence as it sounds like in the section on loneliness (not only an issue there)? It is important to make that very clear whether women refer to past violence, potentially 20 years ago or current violence. For example, but not only there, in the section “Silence, stigma and family” it is unclear if the women refer to the current violence they experienced or past violence, potentially 20 years ago. The whole paragraph seems to refer to varies time points in which the violence happened and this needs to be clarified, especially since there is a specific sub-section for it

We appreciate this important comment, and recognize that throughout the evidence included in this review, there are manuscripts that focus on previous and current violence or current violence only, and that in some cases, given women’s experiences of violence as continuous and interlinked throughout their lifetimes, the distinction is difficult to identify. We have included references to this issue within the Results section where relevant, such as:

“Shifts in types of violence, from predominantly physical violence to predominant psychological abuse and neglect, were commonly described in studies that encompassed previous and on-going IPV. Studies focused on IPV more commonly described both previous and on-going violence, and a small number described only violence experienced while aged 50 or above.”

In the section on Suffering, loneliness, regret and guilt:

“The studies that explored these themes primarily encompassed accounts of violence experienced throughout intimate relationships – while women were younger and through to older age. These experiences were described and conceptualized by older women as interlinked and continuity of victimization by intimate partners was emphasized, rather than viewing women’s experiences of violence in older age as distinct or separate.”

In the section on Violence, ageing and vulnerabilities: 

“These studies primarily focused on current experiences of violence of older women, as changes in physical and emotional capacity to cope was described in relation to present victimization.”

In the section on Silence, stigma and family:

These studies primarily focused on previous and current experiences of violence; social norms predominant when women were younger shaped prior and current responses.”

4. Have you actually found any differing evidence by different age categories among women aged 50 or above? In the limitation section you mention that the age rages varied widely, but it would be could to situate the results into this as 50 to 64 is quite a different age category than 65 to 99, when women are also more likely to be retired.

We found this to be one of the major limitations of the available evidence, and have added a sentence to make this limitation clearer based on this comment:

“The available evidence does not disaggregate findings to enable understanding of whether or how women in different age groups experience violence differently, despite significant variation in living conditions, employment and health status of women aged 50-64 vs. 65 and up, for example.”

5. I am a bit surprised by the structuring of the results section and consider reworking it as it jumps from the overview of forms of violence and perpetrators in a summary paragraph to causes, consequences, risk factors to financial abuse- a form of violence experienced, norms, needs, early childhood violence as a risk factor later. While the overall heading make sense, the sub heading sometimes seem to belong somewhere else and do not flow.

We agree with this comment, which is also in line with Reviewer 2’s feedback regarding structure of the Results section. As such, we have refined our coding structure and made the following changes:

• Substantially expanded on the first section of results, renaming if Descriptions and patterns of types of violence, including integrating the sub-theme on increases of violence into this section;

• Removing the section on causes of elder abuse (as we do not have separate sections on causes of other specific types of abuse);

• Creating a theme on issues relating to the perpetrator, and including the sub-themes Ageing Perpetrators and continuity of abuse and

• Removed Ageing and changes in the nature and patterns of IPV theme (sub-themes moved to Descriptions and patterns of types of violence and Perpetrator related factors)

• Removed Control and financial abuse among older women as a sub-theme and incorporated parts of it into Descriptions and patterns of types of violence

6. Did you actually find different results by LMIC and HIC?

Given the very small number of studies conducted in LMIC, this was not possible and we have added a sentence in the Limitations section to indicate this:

“In addition, the small number of studies conducted in low and middle-income countries entailed that comparison of patterns between high-income and low and middle-income contexts was not possible.”

7. The section “Descriptions of types of violence and perpetrators” is actually a quite crucial one, but it is currently very descriptive in terms of numbers. Given that forms of violence and perpetrators are such a key objective, could you expand this section and show what kind of violence was perpetrated by the different perpetrator types or forms of abuse and whether there were any age trends among the older women.

We appreciate this helpful comment, and have substantially expanded this section. We have renamed it Description and patterns of types of violence, and have a separate section focusing more on perpetrators. The section now reads:

Older women described IPV, family violence and elder abuse of various types, perpetrated by a range of perpetrators [Table 1]. Among the specific types of violence reported in the articles in this review, across IPV, elder abuse and family violence, physical violence was most frequently reported [23, 28-50, 53, 56-59, 62, 65-75], followed by emotional/ psychological [24, 28, 32-35, 37-47, 49, 50, 52, 56-58, 62-75], economic/ financial [30-32, 35, 37, 39-41, 44-46, 57, 58, 60, 64, 67, 68, 70-73], sexual [23, 29, 30, 36, 38, 45, 46, 50, 53, 55, 56, 63, 68, 70, 71, 75], verbal [28, 36, 41, 48, 58, 64-66, 69, 72], controlling behaviors [41, 44, 45, 47, 49, 60, 63, 66, 72], and lastly, neglect [24, 31, 35, 57, 58, 64, 67].

Older women’s experience of IPV was the most frequent form of violence reported (41 articles) [23, 25, 29, 30, 32-34, 36-38, 41-52, 55-61, 63-75]. Older women described on-going instances of neglect, verbal abuse and financial exploitation in a study conducted in India; in other cases, physical violence characterized earlier and on-going experiences of violence within intimate partner relationships. IPV in particular was described by older women as occurring throughout different stages in the relationship, spanning their youth and into older age. Older women often experienced an escalation of IPV and controlling behaviors despite the age and/ or illness of their partner [32, 36, 42, 57, 65, 72]. Changing relationship dynamics due to ageing – including a husband’s retirement, children leaving the home, women wanting to engage in activities outside of the home, or diagnosis of a chronic or terminal illness – triggered escalating IPV [32, 36, 42, 43, 52, 65]. Shifts in types of violence, from predominantly physical violence to predominant psychological abuse and neglect, were commonly described in studies that encompassed previous and on-going IPV. Studies focused on IPV commonly described both previous and on-going violence, and a smaller number described only or primarily violence experienced while aged 50 or above.

Violence occurring within the family was discussed in 14 articles [31, 35, 39, 40, 42, 44, 45, 51, 59, 62, 64, 65, 67, 74], with perpetrators including family members not including children [39, 40, 44, 45, 59, 64, 65, 67] and adult children [31, 35, 44, 62, 74]. Studies captured instances of physical and verbal aggression by mentally ill adult children against older women, violence enacted by elderly with dementia against older women who were caregivers and forms of neglect, financial exploitation and emotional abuse enacted by family members, including children. The majority of these studies focused on violence experienced in older age, while one study explored dynamics of abuse between children and mothers across the lifespan. 

10 articles reported on experiences of elder abuse [24, 28, 31, 39, 40, 51, 53-55, 73] with perpetrators including community members [39, 40], caregivers [53, 55], nursing home residents [54] and health care providers [24]. Types of elder abuse included verbal abuse, physical assault and inappropriate sexual advances and sexual assault.

Financial control spanned instances of elder abuse, family violence and IPV [39, 49, 58, 60, 73], and was described as co-occurring with and resulting in other forms of violence. Financial exploitation could result in emotional and/ or physical violence if older women resisted control [58, 60]. An older woman explained that in the context of her relationship with her husband, “If I did not follow his control [over money], he would be verbally abusive” [60].

8. Page 4, line 77 needs a reference

We have added a reference. 

9. Page 6, line 116: Can you state if these systematic reviews said anything about the perpetrators of this violence? Which age ranges did the systematic reviews investigate?

We have clarified analysis of perpetrators within these systematic reviews and age ranges investigated. This section now reads:

“Employing an older adult mistreatment framework, a systematic review of quantitative studies of elder abuse (against men and women aged 60+) found that the global prevalence of elder abuse in community settings is 15.7% in the past year, with psychological abuse and financial abuse as the most prevalent forms of abuse reported [15]. This review reported prevalence by type of violence, but did not report on perpetrators. Analysis of studies conducted in institutional settings found women, aged 60 and above, to be significantly more vulnerable to abuse, with psychological abuse as the most prevalent form of violence, followed by physical violence, neglect, financial and sexual abuse [16]; this analysis included data reporting staff-to-resident abuse. Analysis of quantitative data of women aged 60 and above in the systematic review of quantitative studies of elder abuse found a global prevalence of elder abuse against women of 14.1% in the past year, with psychological abuse reported as the most prevalent form of violence, followed by neglect. The focus of this review was prevalence of different sub-types of violence, and type of perpetrator was not considered. Another systematic review of quantitative data on interpersonal violence (physical and/or sexual violence) against older women (aged 65 and above) in community dwellings primarily employed an IPV framework, finding prevalence of reported interpersonal violence ranged from 6 to 59% over a lifetime, from 6 to 18% since turning 50, and 0.8 to 11% in the past year, however, results indicated that definitions of violence vary widely and affect prevalence estimates [17]. Syntheses of quantitative literature have identified prevalent forms of violence against older women, highlighting limitations in the evidence-base due to variations in definitions and methodology, and a primary emphasis on populations in high-income, Western countries. These reviews have captured a wide range of types of violence, however, have not considered type of perpetrators or patterns of co-occurring types of violence.” 

10. Did you use any time limit for the search?

No, we did not use and time limit for the search. We have clarified this in the methods section.

11. Did forms/types of violence cluster in certain countries?

The studies were primarily conducted in USA and Israel, followed by UK and Canada. While there are some patterns identified within those countries, we concluded that we could not adequately assess whether this reflected true patterns in violence against older women, or the research focus of the research teams in these countries.

12. Results: On page 14 under the heading “Descriptions of types of violence and perpetrators” you first describe the terms IPV, family violence and elder abuse, but these terms have not been described before in terms of what they mean and how they are different from each other

We agree that the manuscript would benefit from definitions of these terms, and have added these definitions into the description of the objectives of the review:

“We include the following forms of violence: elder abuse, family violence and intimate partner violence. Elder abuse is defined as “single or repeated act, or lack of appropriate action, occurring within any relationship where there is an expectation of trust which causes harm or distress to an older person” Intimate partner violence is defined as “behaviour by an intimate partner or ex-partner that causes physical, sexual or psychological harm, including physical aggression, sexual coercion, psychological abuse and controlling behaviours.” Family violence is often used interchangeable with intimate partner violence, however, also encompasses abuse and violence perpetrated by other family members, for example, adult children or in-laws.”

13. On page 15 when you talk about causes of violence, which types of violence were referred to?

We have removed the theme Causes of Elder Abuse, given we do not have separate themes for causes of other types of violence analysed. We incorporated the analysis regarding ageist attitudes as a cause of violence into the theme, Social and gender norms regarding response to violence; Sub-theme, Perceptions of abuse and violence as normal. 

14. Is the financial abuse section actually referring to tis as a form of violence or a cause for other violence or a co-occurrence of numerous forms of violence?

We agree that in its previous format the inclusion of Financial Abuse as a separate section was not clear enough. As such, we have incorporated aspects of this sub-theme into the Description and patterns of types of violence section, to add further detail there and ensure inclusion of this important type of violence, while improving the structure and flow of the Results. 

15. In the discussion section you refer to health care providers response and women’s concerns regarding confidentiality, however, this was not brought up in the results at all and should have been mentioned there too.

We have included the following text in the Results section relating to health car providers’ response:

“Older women reported several concerns when interacting with health care providers, including health care providers’ assumptions that older women could not be experiencing violence due to their age, minimization of forms of abuse common to older women, and lack of confidentiality when using the same provider as their spouse [70].”

16. Implication for future research – the scarcity of research only relates to qualitative work or more generally? I understand that your review mainly found studies on IPV, but what did the quantitative reviews find? Would they support your claim?

The quantitative evidence-base includes a systematic review of elder abuse against older women, and much more literature encompassing different forms of violence against older women. We have added the following text to indicate this:

“In the quantitative evidence-base, systematic reviews have focused on elder abuse and on IPV. A review of quantitative evidence on IPV identified 19 studies and the review of elder abuse against women included 50 studies; as such, the quantitative evidence-base appears to capture more in terms of forms of violence against older women.”

---

## [Decision Letter · Decision Letter 1]

9 Sep 2020

Violence against older women: a systematic review of qualitative literature

PONE-D-20-10316R1

Dear Dr. Meyer,

We’re pleased to inform you that your manuscript has been judged scientifically suitable for publication and will be formally accepted for publication once it meets all outstanding technical requirements.

Kind regards,

Stefano Federici, Ph.D.

Academic Editor

PLOS ONE

Additional Editor Comments (optional):

Reviewers' comments:

Reviewer's Responses to Questions

**Comments to the Author**

1. If the authors have adequately addressed your comments raised in a previous round of review and you feel that this manuscript is now acceptable for publication, you may indicate that here to bypass the “Comments to the Author” section, enter your conflict of interest statement in the “Confidential to Editor” section, and submit your "Accept" recommendation.

Reviewer #1: All comments have been addressed

Reviewer #2: All comments have been addressed

Reviewer #3: All comments have been addressed

2. Is the manuscript technically sound, and do the data support the conclusions?

Reviewer #1: (No Response)

Reviewer #2: (No Response)

Reviewer #3: Yes

3. Has the statistical analysis been performed appropriately and rigorously? 

Reviewer #1: (No Response)

Reviewer #2: (No Response)

Reviewer #3: N/A

4. Have the authors made all data underlying the findings in their manuscript fully available?

Reviewer #1: (No Response)

Reviewer #2: (No Response)

Reviewer #3: Yes

5. Is the manuscript presented in an intelligible fashion and written in standard English?

Reviewer #1: (No Response)

Reviewer #2: (No Response)

Reviewer #3: Yes

6. Review Comments to the Author

Reviewer #1: The author responded to suggestions in my first review of this manuscript and I have nothing further to add. I am comfortable with this manuscript moving to publication.

Reviewer #2: (No Response)

Reviewer #3: Thank you for addressing all my comments in such a diligent way. The article is a great contribution to the field! Thank you for all your hard work on it

7. PLOS authors have the option to publish the peer review history of their article (what does this mean?). If published, this will include your full peer review and any attached files.

Reviewer #1: No

Reviewer #2: No

Reviewer #3: **Yes: **Heidi Stöckl

---

## [Editor Report · Acceptance letter]

14 Sep 2020

PONE-D-20-10316R1

Violence against older women: a systematic review of qualitative literature

Dear Dr. Meyer:

I'm pleased to inform you that your manuscript has been deemed suitable for publication in PLOS ONE. Congratulations! Your manuscript is now with our production department.

Kind regards,

on behalf of

Prof. Stefano Federici 

Academic Editor

PLOS ONE